# Molecular, cellular, and developmental organization of the mouse vomeronasal organ at single cell resolution

Max Henry Hills[1], Limei Ma[1], Ai Fang[1], Thelma Chiremba[1], Seth Malloy[1], Allison R Scott[1], Anoja G Perera[1], C Ron Yu[1,2]*

[1]Stowers Institute for Medical Research, Kansas City, United States; [2]Department of Cell Biology and Physiology, University of Kansas Medical Center, Kansas City, United States

## eLife Assessment

The manuscript by Hills, et al. presents data that support multiple conclusions regarding the gene expression patterns of cells, especially chemosensory neurons. The evidence is largely **solid**, with transcriptomic analysis combined and validated by spatially resolved expression in tissue sections, but is incomplete in other ways with some claims not fully supported. This large-scale single-cell transcriptomics dataset is an **important** resource, alongside a thorough exploration of the molecular features of the different cell types within the mouse vomeronasal organ, including expression of chemosensory receptors.

**Abstract** We have generated single cell transcriptomic atlases of vomeronasal organs (VNO) from juvenile and adult mice. Combined with spatial molecular imaging, we uncover a distinct, previously unidentified class of cells that express the vomeronasal receptors (VRs) and a population of canonical olfactory sensory neurons in the VNO. High-resolution trajectory and cluster analyses reveal the lineage relationship, spatial distribution of cell types, and a putative cascade of molecular events that specify the V1r, V2r, and OR lineages from a common stem cell population. The expression of vomeronasal and olfactory receptors follow power law distributions, but there is high variability in average expression levels between individual receptor and cell types. Substantial co-expression is found between receptors across clades, from different classes, and between olfactory and VRs, with nearly half from pairs located on the same chromosome. Interestingly, the expression of V2r, but not V1r, genes is associated with various transcription factors, suggesting distinct mechanisms of receptor choice associated with the two cell types. We identify association between transcription factors, surface axon guidance molecules, and individual VRs, thereby uncovering a molecular code that guides the specification of the vomeronasal circuitry. Our study provides a wealth of data on the development and organization of the accessory olfactory system at both cellular and molecular levels to enable a deeper understanding of vomeronasal system function.

## Introduction

In many terrestrial species, the vomeronasal organ (VNO) is dedicated to the detection of inter- and intra-species chemosensory cues (*Birch, 1974*; *Wyatt, 2003*; *Brennan and Zufall, 2006*; *Isogai et al., 2011*; *He et al., 2008*). Detection of these cues triggers innate neuroendocrine responses and elicits stereotypic social and reproductive behaviors (*Birch, 1974*; *Wyatt, 2003*; *Bruce, 1969*; *Drickamer and Assmann, 1981*; *Halpern and Martínez-Marcos, 2003*; *Vandenbergh, 1983*; *Vandenbergh,*

*For correspondence:
cry@stowers.org

Competing interest: The authors declare that no competing interests exist.

*1989*; *Clancy et al., 1984*; *Maruniak et al., 1986*; *Meredith, 1998*; *Lonstein and Gammie, 2002*; *Ferguson et al., 2002*; *Dey et al., 2015*). The VNO shares a developmental origin with the main olfactory epithelium (MOE), which detects the odor world at large and allows associative learning to take place. Both develop from the olfactory placode during early embryogenesis (*Yoshida et al., 1993*), but the two sensory organs follow different developmental trajectories to establish distinct characteristics in morphology, cellular composition, and molecular features. Single cell atlases of the MOE have been generated to reveal astonishing details in its molecular composition and developmental trajectories (*Olender et al., 2016*; *Tsukahara et al., 2021*; *Hanchate et al., 2015*; *Fletcher et al., 2017*; *Wu et al., 2018*). Transcriptomic data of the VNO is not extensive (*Villamayor et al., 2021*; *Duyck et al., 2017*), but single cell analyses have already provided critical information about VNO development (*Katreddi et al., 2022*; *Lin et al., 2022*). In this study, we generate single cell atlases of developing and adult mouse VNO neuroepithelia to answer fundamental questions about the VNO.

The vomeronasal sensory neurons (VSNs) express three families of G-protein-coupled receptors, the V1rs, V2rs, and the formyl peptide receptors (Fprs) (*Herrada and Dulac, 1997*; *Ryba and Tirindelli, 1997*; *Rivière et al., 2009*; *Liberles et al., 2009*; *Dulac and Axel, 1995*; *Matsunami and Buck, 1997*; *Rodriguez et al., 2002*). With more than 400 members, the vomeronasal receptors (VRs) are among the fastest evolving genes (*Grus and Zhang, 2006*; *Shi and Zhang, 2007*; *Silva and Antunes, 2017*; *Lane et al., 2004*; *Kurzweil et al., 2009*; *Zhang et al., 2004*). Signaling pathways in the VNO include the Gi2 and Go proteins, and a combination of *Trpc2*, *Girk1*, *Sk3*, and *Tmem16a* ion channels, to transduce activation of the VRs (*Liman and Corey, 1996*; *Yu, 2015*; *Wu et al., 1996*; *Dibattista et al., 2008*; *Dibattista et al., 2012*; *Amjad et al., 2015*; *Zhang et al., 2010*; *Yang and Delay, 2010*; *Spehr et al., 2002*; *Lucas et al., 2003*; *Kelliher et al., 2006*; *Berghard and Buck, 1996*; *Kim et al., 2012*; *Kim et al., 2011*; *Trouillet et al., 2019*; *Chamero et al., 2011*; *Menco et al., 2001*; *Zufall, 2005*; *Stowers et al., 2002*; *Leypold et al., 2002*; *Liman and Dulac, 2007*). The spatially segregated expression of *Gi2* and *Go*, as well as the VR genes suggests two major classes of neurons. Our study reveals new classes of sensory neurons, including a class of canonical olfactory sensory neurons (OSNs) in the VNO. We further determine the developmental trajectories of the separate lineages and the transcriptional events that specify these lineages.

Pheromones are highly specific in activating the VSNs (*He et al., 2008*; *Leinders-Zufall et al., 2000*; *Haga-Yamanaka et al., 2015*; *He et al., 2010*; *Boschat et al., 2002*; *Holy et al., 2000*; *Nodari et al., 2008*). Previous studies have shown that VSNs residing in the apical layer of the VNO express V1R (Vmn1r) genes in a monoallelic manner (*Boschat et al., 2002*; *Roppolo et al., 2007*; *Rodriguez et al., 1999*), whereas the basal VSNs express one or two of the broadly expressed C clade of V2R (Vmn2r) genes, plus a unique V2R gene belonging to another clade (*Herrada and Dulac, 1997*; *Ryba and Tirindelli, 1997*; *Rivière et al., 2009*; *Liberles et al., 2009*; *Dulac and Axel, 1995*; *Matsunami and Buck, 1997*; *Martini et al., 2001*; *Silvotti et al., 2011*; *Silvotti et al., 2007*). Although these results suggest that receptor expression in the apical VNO conforms to the 'one neuron one receptor' pattern as found in the MOE, the mechanisms that control receptor expression are unknown. Here, we find substantial co-expression of VRs, and of vomeronasal and odorant receptors. Moreover, our analyses indicate that selection of V1R expression likely results from stochastic regulation as in the OSNs, but V2R expression likely result from deterministic regulation.

Finally, we address the molecular underpinning of how VSNs establish anatomical connections to transmit sensory information. VSNs expressing the same receptor project to dozens of glomeruli in the AOB (*Rodriguez et al., 1999*; *Belluscio et al., 1999*). Individual stimuli activate broad areas in the AOB (*Meeks et al., 2010*). Moreover, the dendrites of the mitral cells in the AOB innervate multiple glomeruli (*Del Punta et al., 2002*; *Wagner et al., 2006*; *Takami and Graziadei, 1991*). This multi-glomerular innervation pattern is in stark contrast with the main olfactory system, where OSNs expressing the same odorant receptor converge their axons into mostly a single glomerulus in each hemisphere of the main olfactory bulb (MOB; *Mombaerts et al., 1996*). The anatomical arrangement in the AOB has strong implications as to how species-specific cues are encoded and how the information is processed. In the MOB, when the convergent glomerular innervation is experimentally perturbed to become divergent, it does not affect detection or discrimination of odorants but diminishes behavioral responses to innately recognized odors (*Gronowitz et al., 2021*; *Qiu et al., 2020*). Thus, stereotypic projection patterns provide a basis for genetically specified connections in the neural circuitry to enable innate behaviors. Consistent with this notion, it has been shown that

mitral cell dendrites innervate glomeruli containing the same VR type such that the divergent projection pattern of the VSNs is rendered convergent by the mitral cells (*Del Punta et al., 2002*). This homotypic convergence suggests that rather than using the spatial position of the glomeruli, the connection between VSNs and mitral cells in the AOB may rely on molecular cues to enable innate, stereotypical responses across different individuals. To a lesser extent, heterotypic convergence, that is, axons expressing different receptors innervating the same glomeruli, is also observed (*Wagner et al., 2006*). In either case, expression of the molecular cues is likely genetically specified and tied to individual VRs, but little is known about how this specification is determined. Our analyses revealed the stereotypic association between transcription factors, axon guidance molecules, and the VRs to suggest a molecular code for circuit specification. Whereas this manuscript highlights some of the main discoveries, much detailed analyses can be found in the dataset hosted online for readers to browse.

## Results

### Cell types in the VNO

We dissected mouse VNOs from postnatal day 14 (P14) juveniles and P56 adults. Cells were dissociated in the presence of actinomycin D to prevent procedure-induced transcription. From four adult (P56) and four juvenile (P14) mice (equal representation of sexes) we obtained sequence reads from 34,519 cells. The samples and replicates were integrated for cell clustering. In two-dimensional UMAP space, 18 cell clusters can be clearly identified (*Figure 1A*). These clusters were curated using known cell markers (*Figure 1B*). There was no obvious difference in the presence of cell clusters between juvenile and adult VNOs (*Figure 1C*) or between male and female sexes (*Figure 1D*). Although there were differences in gene expression profiles between the ages (*Figure 1—figure supplement 1*) and sexes (*Figure 1—figure supplement 2*), the list of significantly differentially expressed genes did not appear to be influential for the neuronal lineage and cell type specification, or related to cell adhesion molecules, which were the main focuses of this study.

Clustering confirmed all previously identified cell types but also revealed some surprises. The largest portion of cells belonged to the neuronal lineage, including the globose basal cells (GBC), immediate neuronal progenitors (INPs), the immature and mature VSNs, and a population of cells that do not co-cluster with either VSN type (see below). There were substantial numbers of sustentacular cells (SCs), horizontal basal cells (HBCs), and ms4-expressing microvillus cells (MVs). Cells engaged in adaptive immune responses, including microglia and T-cells, were also detected. A population of Fpr-1 expressing cells that were distinctive from the VSNs expressing the Fpr family of genes formed a separate class. These were likely resident cells mediating innate immune responses. We also identified the olfactory ensheathing cells (OECs) and a population of lamina propria (LP) cells, which share molecular characteristics with what we have found in the MOE (*Wu et al., 2022*).

To obtain the spatial location of the various cell types, we selected 100 target genes based on the scRNA-seq results. Using probes for these genes, we used the Molecular Cartography platform to perform spatial molecular imaging (*Figure 1E*). Based on molecular clouds and DAPI nuclei staining, we segmented the cells and quantified gene expression profiles to cluster the cells. We then map individual cell clusters onto their spatial locations in VNO slices. Unlike previous studies that relied on a few markers to identify cell types, our approach relied on the spatial transcriptome to calculate the probability that a cell belongs to a specific class. This analysis revealed that the VSNs and supporting cells are located in the pseudostratified neuroepithelium (*Figure 1F*). The LP cells are located along the LP underlying the neuroepithelium as found in the MOE (*Figure 1F and G*). Surprisingly, however, there are few HBCs located along the basal lamina, in direct contrast to the MOE (*Figure 1F and G*). Most of the HBCs are found in the non-neuronal epithelium surrounding the blood vessel, with few near the marginal zone. The marginal zone is thought to be the neurogenic region (*Takami and Graziadei, 1991*; *Mombaerts et al., 1996*; *Gronowitz et al., 2021*). We quantified the distribution of cell types in various regions of the VNO neuroepithelia (*Figure 1—figure supplement 3A*) and found significantly more GBCs, INPs, and immature VSNs in the marginal zone than in the main zone (*Figure 1—figure supplement 3B* and *Figure 1—figure supplement 3C*). A previous study suggested neurogenic activity in the medial zone (*Brann and Firestein, 2010*), but we did not find evidence in support of that conclusion, which was largely based on BrdU staining of mitotic cells

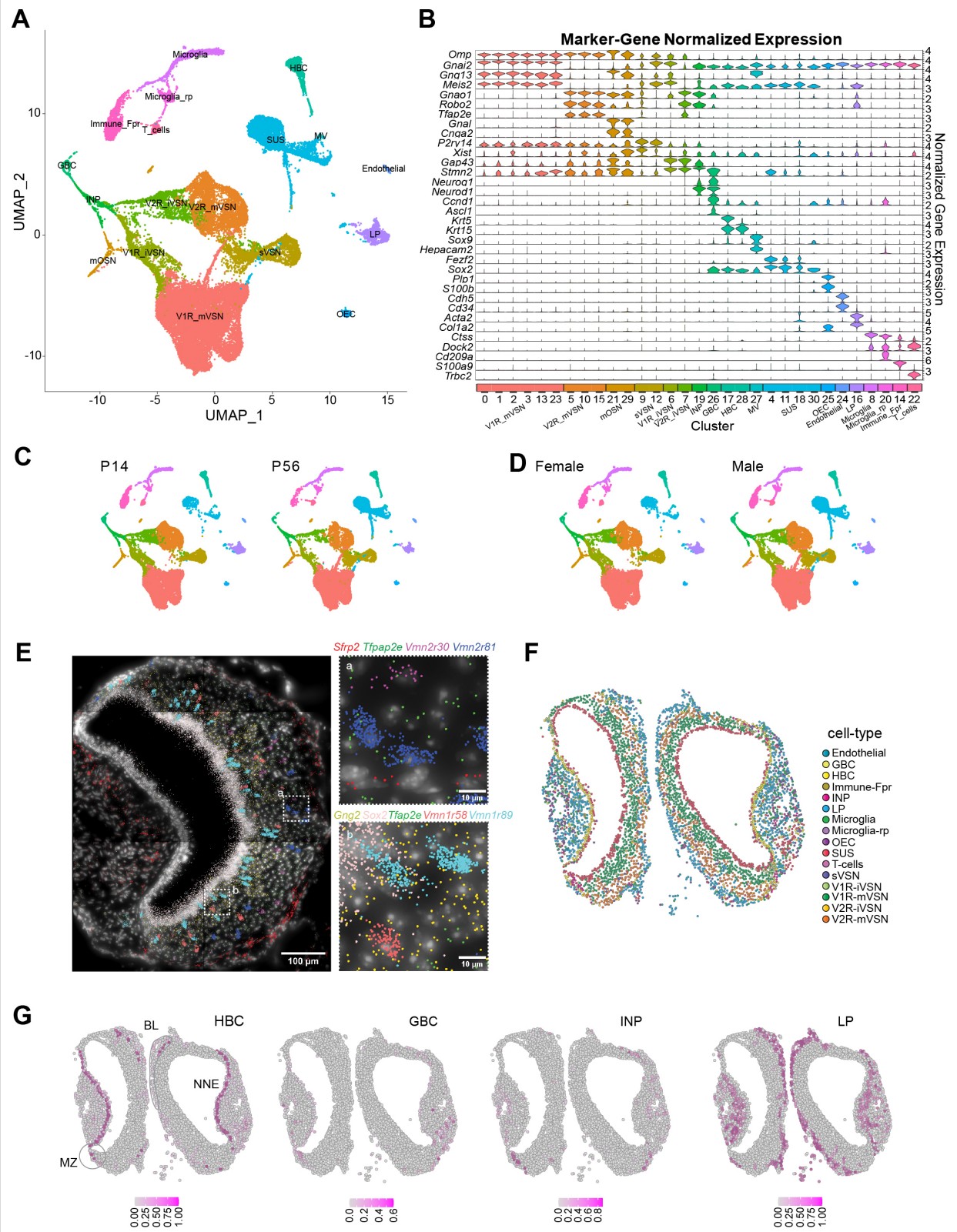

**Figure 1.** Single cell transcriptomic profile of the whole vomeronasal organ. UMAP visualization of integrated cell-type clusters for whole-VNO single-cell RNA-seq. (**B**) Cell-type marker-gene normalized expression across the cell clusters. C. UMAP of cell-type clusters split by age. (**D**) UMAP of cell-type clusters split by sex. (**E**) A representative image of transcript distribution for 9 genes in a VNO slice using the Molecular Cartography platform Resolve Biosciences. Insets (**a and b**) show the magnified image of areas identified in the main panel. Individual cell shapes can be determined from the

*Figure 1 continued on next page*

*Figure 1 continued*

transcript clouds. (**F**) Spatial location of individual VNO cells color-coded according to cell type prediction based on the spatial transcriptomic analysis. (**G**) Location of cell belonging to HBC, GBC, INP, and LP cell types, respectively. Heat indicates confidence of predicted values. BL: basal lamina; MZ: marginal zone.

The online version of this article includes the following figure supplement(s) for figure 1:

**Figure supplement 1.** Age differences in gene expression.

**Figure supplement 2.** Sex differences in gene expression.

**Figure supplement 3.** Zonal distribution of cell types in the VNO neuroepithelia.

without lineage-specific information. Based on our transcriptomic analysis, we conclude that neurogenic activity is restricted to the marginal zone.

## Novel classes of sensory neurons in the VNO

To better understand the developmental trajectory of the VSNs, we segregated the cells in the neuronal lineage from the whole dataset (*Figure 2A*). The neuronal lineage consists of the GBCs, INPs, immature neurons as determined by the expression of *Gap43* and *Stmn2* genes (*Figure 2—figure supplement 1A and B*), and the mature VSNs. Cells expressing *Xist*, which is expressed in female cells, were intermingled with those from males (*Figure 2—figure supplement 1C*). This observation is consistent with our previous studies using bulk sequencing indicating that the cell types are not sexually dimorphic (*Duyck et al., 2017*). It is also consistent with physiological responses of the VSNs to various stimuli (*He et al., 2008*; *He et al., 2010*; *Holy et al., 2000*; *Tolokh et al., 2013*; *Arnson et al., 2010*). For further analyses, therefore, we did not segregate the samples according to sex.

The VSNs are clearly separated into the V1R and V2R clusters as distinguished by the expression of *Gnai2* and *Gnao1*, respectively (*Figure 2B*). Surprisingly, we observed that a portion of the V2R cells also expressed *Gnai2*. These cells were primarily from adult, but not juvenile male mice (*Figure 2—figure supplement 1D*). Past studies based on *Gnai2* and *Gnao1* expression would have identified this group of cells as V1R VSNs, but the transcriptome-based classification places them as V2R VSNs. The signaling mechanism of this group of cells may be different from the canonical V2R VSNs.

We found a major group of cells that expressed the prototypical VSN markers but formed a cluster distinct from the mature V1R and V2R cells (*Figures 1A and 2A*). Within the cluster, there was an apparent segregation among the cells into V1R and V2R groups based on marker gene expression (*Figure 2B*). A small group of OR-expressing neurons also belonged to this cluster. These cells expressed fewer overall genes and with lower total counts when compared with the mature V1R and V2R cells (*Figure 2—figure supplement 1E and F*). The lower ribosomal gene expression suggests that these neurons are less active in protein translation (*Figure 2—figure supplement 1G*). This group of cells are scattered within the epithelium (*Figure 2E*).

There are 503 differentially expressed genes in this cluster when compared with other mature neurons (p*adj* <0.001; FC >1.5; *Figure 2F*, *Figure 2—figure supplement 1H*). Gene ontology (GO) analysis reveals that differentially expressed genes enriched in this group are associated with odorant binding proteins (*Figure 2G*). Correspondingly, mucin 2 (*Muc2*), odorant binding protein 2 a (*Obp2a*), *Obp2b*, and lipocalin 3 (*Lcn3*) genes, usually enriched in non-neuronal supporting or secretory gland cells, are expressed at higher levels in these neurons than the canonical VSNs (*Figure 2H* and *Figure 2—figure supplement 2A–D*). They also have higher expression of *Wnk1*, which is involved in ion transport, the lncRNA *Neat1*, the purinergic receptor *P2ry14*, the zinc finger protein *Zfp738*, and the centrosome and spindle pole associated *Cspp1* (*Figure 2—figure supplement 2E–I*). On the other hand, these cells exhibit lower expression of *Omp* and *JunD* (*Figure 2F* and *Figure 2—figure supplement 2M–N*). The lower level of *Omp* suggests that these cells do not have the full characteristics of mature VSNs (mVSNs), but they are also distinct from the immature VSNs (iVSNs) as they do not express higher levels of immature markers such as *Gap43* and *Stmn2* (*Figure 2—figure supplement 1A and B*). The lower expression of *Ndua2*, *Ndua7*, *Cox6a1*, and *Cox7a2*, which are involved in mitochondrial activities, suggest that these cells are less metabolically active than the canonical VSNs (*Figure 2F* and *Figure 2—figure supplement 2O–R*). To rule out the possibility that these cells were immature or senescent, we performed a pseudotime analysis on the neuronal lineage and found the cluster to have similar pseudotime values as the mature V1R and V2R linages (*Figure 2—figure*

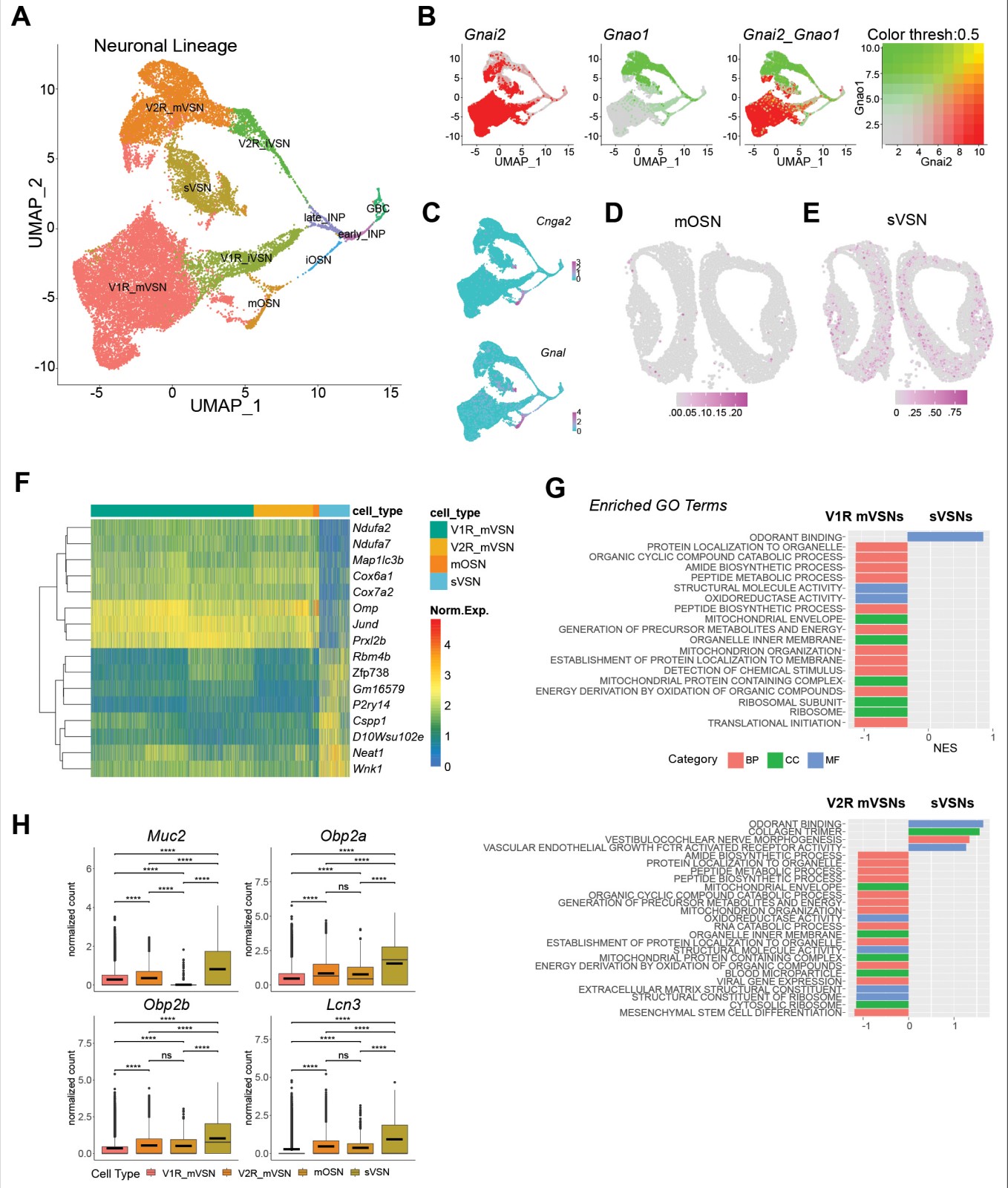

**Figure 2.** Novel neuronal lineage. (**A**) UMAP visualization of cell-type clusters for the neuronal lineage. (**B**) Expression of Gnai2 and Gnao1 in the neuronal lineage. (**C**) Expression of Cnga2 and Gnal in the OSN lineage. (**D–E**) Location of mOSNs (**D**) and sVSNs (**E**) in a VNO slice. Color indicates prediction confidence. (**F**) Heatmap of normalized expression for a select set of mutually differentially expressed genes between sVSNs and mature V1Rs, V2Rs, and OSNs. (**G**) Enriched gene ontology (GO) terms in sVSNs when compared with V1R and V2R VSNs, respectively (GSEA Permutation

*Figure 2 continued on next page*

*Figure 2 continued*

testing w/ FDR ≤ 0.05). (**H**) Box plots of normalized expression for Muc2, Obp2a, Obp2b, and Lcn3 across mature sensory neurons (Wilcoxon rank sum test, FDR ≤ 0.05).

The online version of this article includes the following figure supplement(s) for figure 2:

**Figure supplement 1.** Neuronal lineage features and differentially expressed genes in sVSNs.

**Figure supplement 2.** Significantly regulated genes in sVSNs.

**Figure supplement 3.** sVSN Pseudotime, OSN markers and GO Terms.

*supplement 3A and B*). Taken together, the gene expression profile suggests that this new class of neurons may not only sense environmental stimulus, but also may provide proteins to facilitate the clearing of the chemicals upon stimulation. We, therefore, name these cells as putative secretory VSNs (sVSNs).

We also identify a distinct set of cells expressing the odorant receptors (ORs) that comprise ~2.3% of the total neurons (*Figure 2A*). A previous study has found the expression of ORs in neurons that display some typical molecular features of VSNs, while another study detecting ORs in the VNO suggested the presence of non-canonical OSNs *Lévai et al., 2006*; *Nakahara et al., 2016*. The OR expressing cells were shown to project to the AOB, but it was not clear how prevalent OR expression was in the VNO, nor whether the cells were VSNs or OSNs. While some OR expressing cells cluster with the mature V1R or V2R neuronal lineages, a majority of these cells lacks V1R or V2R markers and forms a cluster distinct from the V1R and V2R VSNs (*Figure 2C*). These cells express *Gnal* and *Cnga2*, which are the canonical markers of OSNs in the MOE.

Differential gene expression analysis (*Figure 2—figure supplement 3C*) reveals an enrichment for multiple GO terms related to cilium (*Figure 2—figure supplement 3D*), consistent with the ciliated nature of OSNs. We thus mark the cells as canonical OSNs. Spatial mapping reveals that the OSNs are mainly in the neuroepithelium, with some cells concentrated in the marginal zone (*Figure 2D*).

## Developmental trajectories of the neuronal lineage

The VNO develops from the olfactory pit during the embryonic period and continues to develop into postnatal stages (*Garrosa et al., 1998*; *Suárez, 2011*; *Katreddi and Forni, 2021*). Neurons regenerate in adult animals (*Giacobini et al., 2000*; *Martínez-Marcos et al., 2000*). Cell types in the vomeronasal lineage have been shown to be specified by BMP and Notch signaling (*Katreddi et al., 2022*; *Naik et al., 2020*), and coordinated by transcription factors (TFs) including Bcl11b/Ctip2, C/EBPγ, ATF5, Gli3, Meis2, and Tfap2e (*Enomoto et al., 2011*; *Lin et al., 2018*; *Nakano et al., 2019*; *Taroc et al., 2020*; *Rawson et al., 2010*; *Chang and Parrilla, 2016*). Recent scRNA-seq studies of the VNO have helped identify Notch signaling as a specifier of the apical and basal lineages and have provided insight into the distinct transcriptional profiles of the basal and apical program (*Katreddi et al., 2022*; *Lin et al., 2022*). Despite these advances in understanding the role of individual genes in VNO development, the transcriptional program that specifies the lineages is not known.

To explore VSN development, we performed pseudotime inference analysis of the V1R and V2R lineages for P14 and P56 mice using *Slingshot* (*Figure 3A*; *Street et al., 2018*). We set GBCs as the starting cluster and mVSNs as terminal clusters. A minimum spanning tree through the centroids of each cluster was calculated using the first fifty principal components and fit with a smooth representation to assign pseudotime values along the principal curve of each lineage. Cell density plots for both the V1R and V2R lineage reveal a higher portion of immature VSNs at P14 than at P56. For the mature VSNs, the P14 samples have peaks at an earlier pseudotime than the P56 mice (*Figure 3B*). This result indicates the VSNs in juvenile mice are developmentally less mature than their counterparts in adults, but these differences do not distinguish them in obvious ways (*Figure 1C*).

On the other hand, we have identified dynamic changes in transcriptomes associated with the V1R and V2R lineages. There were 2037 significantly differentially expressed genes between V1R and V2R lineages (*Figure 3C*). While V1R and V2R lineages shared some of the genes in the early stages of development, most show distinct expression patterns between the two. To determine the transcriptional program associated with the lineages, we examined the expression sequence of individual transcription factors and signaling molecules from the gene list.

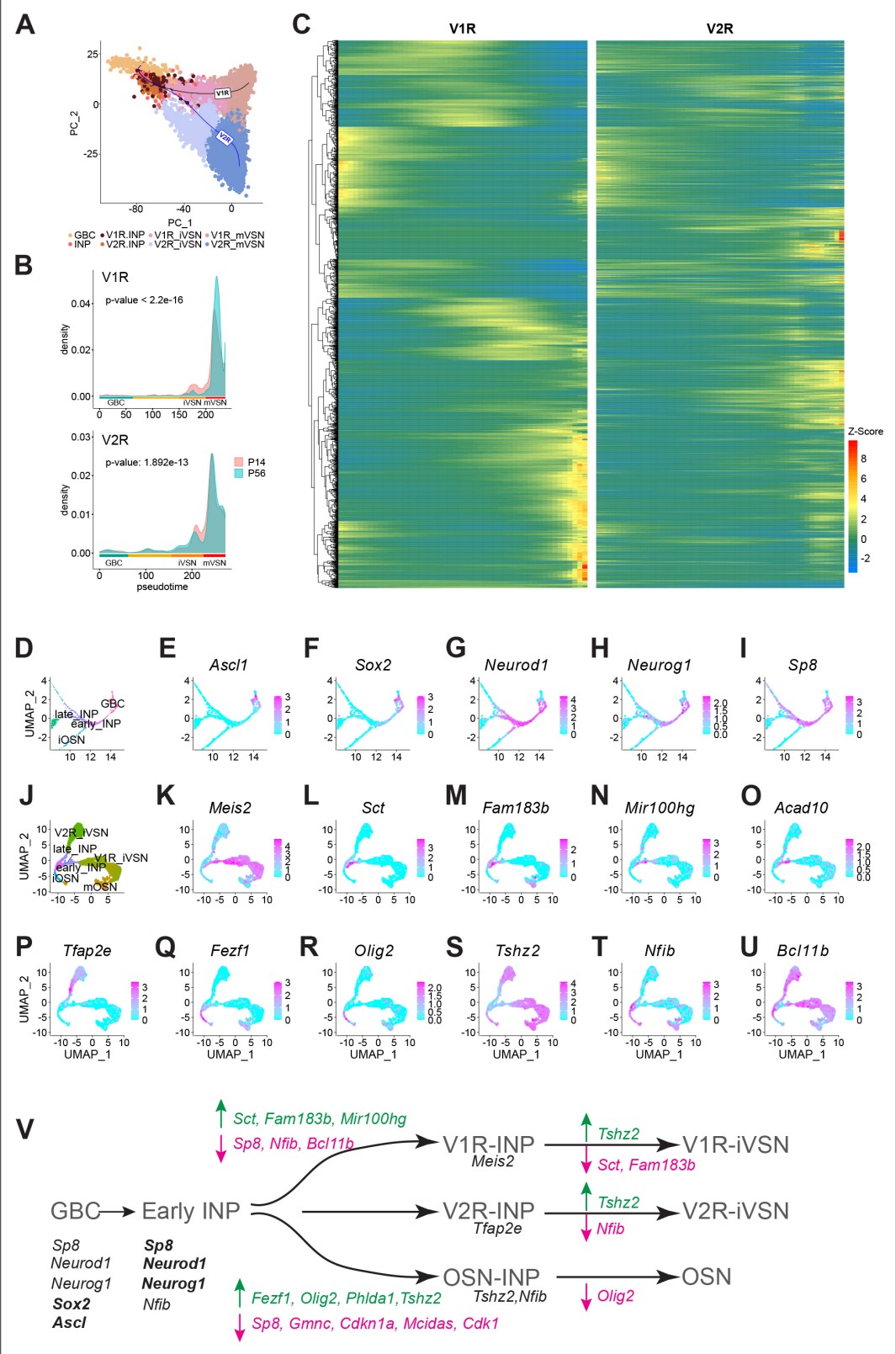

**Figure 3.** Molecular cascades separating the neuronal lineages. (**A**) PCA plot of V1R and V2R pseudotime principal curves across cell types. (**B**) Cell density plots across pseudotime for P14 and P56 mice (Two-sided Kolmogorov-Smirnov test). (**C**) Heatmap showing expression in pseudotime for genes that differentially expressed between the V1R and V2R lineages. Heat indicates Z-score values. (**D**) Zoomed in UMAP of cell types early in the neuronal

*Figure 3 continued on next page*

*Figure 3 continued*

lineage. (**E–I**) Feature plots for select genes expressed early in the neuronal lineage. (**J**) UMAP of INP, iVSN, iOSN, and mOSN cell types. (**K–U**) Normalized expression of candidate genes for V1R/V2R/OSN lineage determination. (**V**) A simplified model for lineage determination by transcription factors in VNO sensory neurons.

The online version of this article includes the following figure supplement(s) for figure 3:

**Figure supplement 1.** Genes expressed during INP to immature neuron transition.

The transcription factor *Ascl1* is expressed by a subset of the GBCs that appears to be the earliest in developmental stage, whereas *Sox2* is expressed by a broader set of GBCs (*Figure 3D–F*). The early INPs are defined by the expression of *Sp8*, *Neurog1,* and *Neurod1*. *Neurog1* is mostly restricted in the early INPs, whereas *Neurod1* and *Sp8* are also expressed by the late INPs (*Figure 3G–I* and *Figure 3—figure supplement 1A–B*). *Neurod1* is expressed by iVSNs and iOSNs, but *Sp8* is only expressed by the iVSNs.

To obtain a refined view of the transition from INPs to the immature sensory neurons, we took the subset and re-clustered them (*Figure 3J*). The late INPs are separated into two clusters that share marker genes with the V1R and V2R iVSNs, respectively, indicating that commitment to the two lineages begins at the late INP stage. Consistent with previous findings, the homeobox protein *Meis2* was specific to the V1R lineage (*Figure 3K*). Concomitant with *Meis2*, there are a number of other genes expressed by the late INPs committed to the V1R fate, including secretin (*Sct*), *Foxj1* target gene *Fam183b*, the microRNA *Mir100hg*, acyl-CoA dehydrogenase *Acad10*, and Keratin7 (*Krt7*; *Figure 3L–O* and *Figure 3—figure supplement 1C*). Notably, *Bcl11b* is exclusively absent from INPs committed to the V1R fate. This observation is consistent with the observation by Katreddi and colleagues that *Bcl11b* is lost in basal INPs in *Notch* knockout (*Katreddi et al., 2022*). Together with the observation that loss of *Bcl11b* results in increased number of V1R VSNs (*Enomoto et al., 2011*), these results indicate that transient downregulation of *Bcl11b* is required for the V1R lineage (*Figure 3U*).

The transcription factor *Tfap2e*, which is required to maintain the V2R VSNs (*Lin et al., 2018*), is expressed by the iVSNs but not by the late INPs committing to the V2R fate (*Figure 3P*). *Sp8* is the only transcription factor specifically expressed in the late INPs committed to the V2R but not the V1R fate (*Figure 3I*). *Emx2* is expressed by all neuronal lineage cells (*Figure 3—figure supplement 1E*). *Krt8* is also found throughout the V2R lineage, but its expression is diminished in the V1R cells (*Figure 3—figure supplement 1D*).

The OSN lineage is marked by the expression of *Olig2*, *Fezf1*, *Tshz2*, and *Nfib* (*Figure 3Q–T*). The expression of *Olig2* and *Fezf1* is exclusive to the OSN fate. *Tshz2* is expressed at a late stage of the iVSNs, but not in the INPs. There are multiple genes that may not directly be engaged in cell fate determination but are clearly markers of the cell types (*Figure 3—figure supplement 1F–R*). Although *Notch1* and *Dll4* are not identified as significantly differentially expressed, feature plots show clear distinction in their expression in the V2R and V1R lineage, respectively, as an earlier study showed (*Katreddi et al., 2022*).

Based on these patterns, we propose a model of molecular cascades that specify the neuronal lineages in the VNO (*Figure 3V*). *Ascl1* and *Sox2* specify the GBCs. The down regulation of *Ascl1*, *Sox2*, and subsequent upregulation of *Neurog1*, *Neurod1*, and *Sox8* commits the cells to the early INPs. The VSN and OSN lineages diverge after the early INP stage. The downregulation of *Sp8*, *Nfib*, and *Bcl11b* and the expression of *Meis2* promote the V1R fate.

The downregulation of *Sp8* and the expression of *Fezf1*, *Tshz2*, and *Olig2* are required for commitment to the OSN lineage. Downregulation of *Sp8* is not required for the V2R lineage, which begin expressing *Tfap2e*. *Neurod1* is expressed in all INPs. These patterns of expression suggest that both the OSN and V1R lineages required the expression of specific transcription factors. The V2R lineage, on the other hand, appears to rely on factors inherited from the early INPs. This suggests the possibility that the V2R lineage is a default path for the VSNs.

## Receptor expression in the VSNs

We quantified the expression of V1Rs, V2Rs, and ORs to gain insights into how chemosensory cues may be encoded by the VNO. For comparison, we also included OR expression from the MOE (*Wu et al., 2022*). Within each class of receptors, the probability of expression of a gene follows a power

law distribution except for the lower ranked genes (*Figure 4—figure supplement 1A*). The sharp deviation from the power law curve for the lower ranked receptors is likely from technical dropout of genes expressed at low levels as they cannot be effectively captured by the scRNA-seq platforms. We plotted the relationship between total reads and the number of cells expressing a given receptor, and the average reads per cell for the receptors (*Figure 4A–H*). We observed weak correlations between the total reads and the cell number expressing a given V1R or a V2R (*Figure 4A and B*). This non-uniform distribution of VR genes is consistent with our observation from bulk sequencing results (*Duyck et al., 2017*).

Most V1Rs and V2Rs are expressed by less than 1500 cells (out of 34,519). Most VRs are expressed at less than 100 counts per cell (*Figure 4E and F*). Several V1Rs, including *Vmn1r184*, *Vmn1r89*, *Vmn1r196*, *Vmn1r43*, and *Vmn1r37*, are highly expressed in individual cells (*Figure 4A*). *Vmn1r196*, *Vmn1r43*, and *Vmn1r37* are also expressed at the highest level per cell. Some others, including *Vmn1r183*, *Vmn1r81*, and *Vmn1r13*, are expressed in large numbers of cells but have low expression in individual cells. Notably, the two highest expressed receptors recognize female pheromone cues. *Vmn1r89*, also known as *V1rj2*, is one of the receptors that detects female estrus signals. *Vmn1r184* is one of the receptors for the female identity pheromone (*Isogai et al., 2011*; *Haga-Yamanaka et al., 2015*; *Haga-Yamanaka et al., 2014*). The functions of *Vmn1r196*, *Vmn1r43*, and *Vmn1r37*, however, are unknown. We did not detect the expression of 45 V1Rs, which may be the result of technical dropout (*Figure 4E*).

We do not observe obvious correlations between expression level and chromosomal locations. Both *Vmn1r89* and *Vmn1r184* are located on Chromosome 7, in a region enriched in V1R genes. Similarly, *Vmn1r37* and *Vmn1r43* are in a V1R-rich region on Chr. 6. However, *Vmn1r183*, *Vmn1r13*, and *Vmn1r81*, which are in the same clusters, are expressed by many cells but at some of the lowest levels.

All V2R genes are detected in the VNO (*Figure 4B and F*). *Vmn2r53*, which has been shown to mediate intermale aggression through a dedicated circuit, has the highest level of expression and is expressed by the second most cells (*Itakura et al., 2022*). Among the highly expressed V2Rs, *Vmn2r1* and *Vmn2r7* are co-receptors for other V2Rs. *Vmn2r59* has been shown to detect predator signals (*Isogai et al., 2011*). *Vmn2r115*, a receptor for ESP22 that is secreted by juveniles (*Ferrero et al., 2013*), is expressed by the highest number of cells. However, *Vmn2r116* (*V2rp5*), which recognizes ESP1 and induces lordosis behavior in females, is a close homolog of *Vmn2r115* but not among the highly expressed genes (*Ferrero et al., 2013*; *Haga et al., 2010*). Notably, *Vmn2r114*, close homolog of *Vmn2r115* and *Vmn2r116*, is also expressed by large numbers of cells. *Vmn2r88*, a hemoglobin receptor (*Osakada et al., 2022*), was not identified as a highly expressed gene. The functions of other highly expressed receptors are not known.

In contrast to the VR genes, total counts for ORs in the VNO exhibit a tight relationship with the number of cells (*Figure 4C and G*). Except for *Olfr124*, most of the OR genes align well with the linear regression curve. This relationship is different from the VR genes and is also distinct from the single cell data from the MOE, which exhibits a similar distribution as the VRs (*Wu et al., 2022*; *Figure 4D and H*). Out of the 686 OR genes detected in the VNO, only 80 are expressed by more than 10 copies per cells, indicating that a majority of the ORs do not contribute to meaningful signaling of chemosensory cues.

To comprehensively survey receptor expression, we also included VR and OR pseudogenes in our analysis (*Figure 4—figure supplement 1B–G*). We did not detect a significant expression of pseudogene V1Rs (*Figure 4—figure supplement 1E*), but *Vmn2r-ps87* has the highest count/cell in the V2R population (*Figure 4—figure supplement 1F*). *Olfr709-ps1*, and *Olfr1372-ps1* are the two highest expressed genes in terms of total count and total number of cells in the VNO (*Figure 4—figure supplement 1D*).

We next examined transcription factors associated with individual receptor types. In the MOE, ORs are monoallelically expressed (*Chess et al., 1994*). The unique expression of an OR gene is mediated by chromosomal repression and de-repression, coordinated by transcription factors and genes involved in epigenetic modification (*Monahan et al., 2019*; *Dalton et al., 2013*; *Clowney et al., 2012*; *Lyons et al., 2013*; *Monahan and Lomvardas, 2015*; *Lomvardas et al., 2006*). Monoallelic V1R gene expression is also observed (*Rodriguez et al., 1999*). Epigenetic modification takes place at V1R gene clusters (*Rodriguez, 2013*), but the repression of receptor genes appears to permit transcriptional stability rather than receptor choice (*Dietschi et al., 2022*). How VR genes are selected is

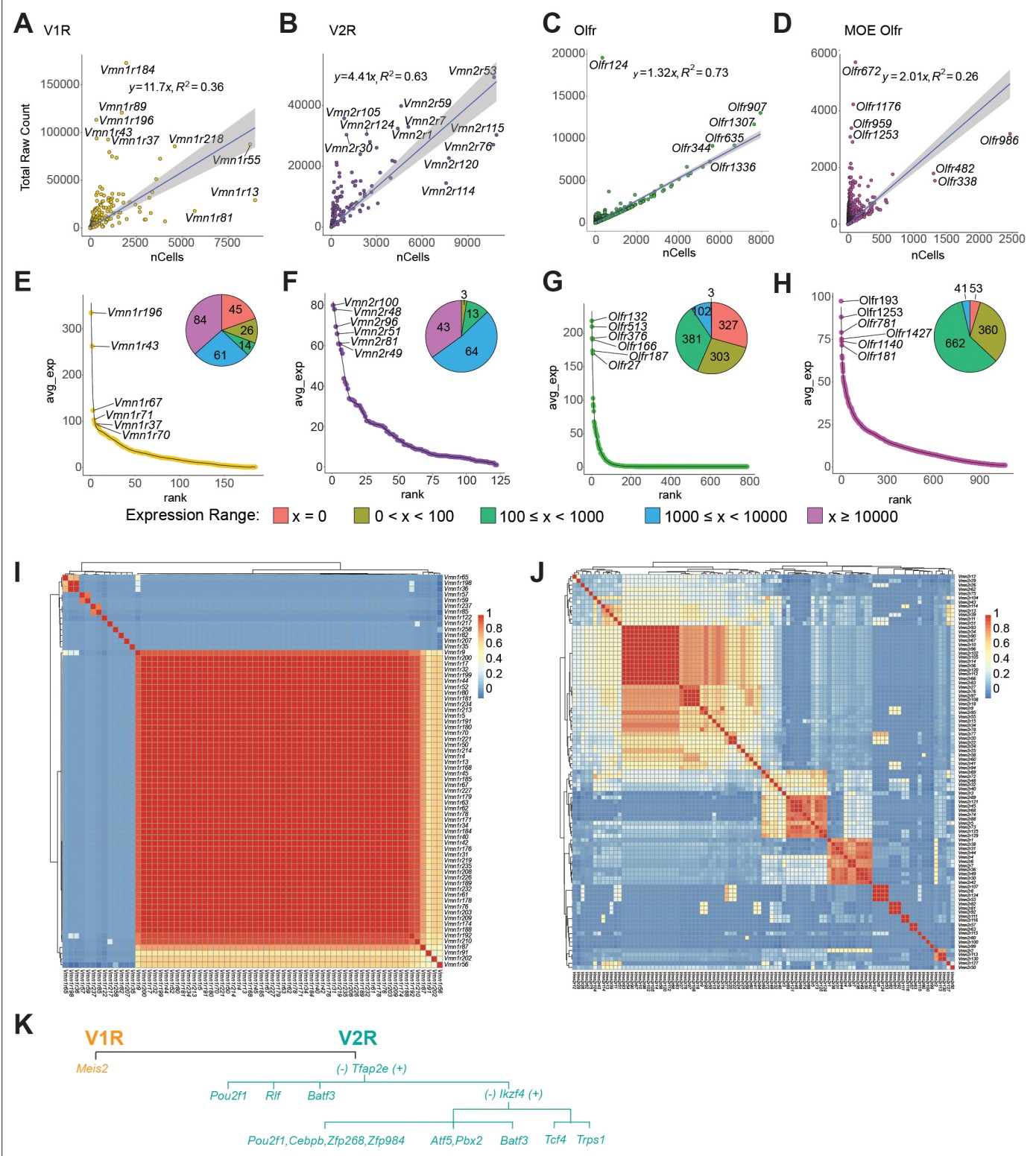

**Figure 4.** Receptor expression in the VNO. (**A-D**) Expression level of individual receptor (total raw counts) plot against the number of cells expressing the receptor for V1R (**A**), V2R (**B**), VNO-Olfr (**C**), and MOE-Olfr (**D**). (**E-H**) Ranked distribution of average expression per cell for the four receptor classes. Inset pie charts show the number of cells expressing a receptor at the specified range. (**I** and **J**) Heatmaps showing the Pearson correlation coefficient of transcription factor expression among the V1Rs (**I**) or V2Rs (**J**). (**K**) A simplified model of transcription factor selection in mVSNs.

The online version of this article includes the following figure supplement(s) for figure 4:

*Figure 4 continued on next page*

**Figure supplement 2.** Co-expression between vomeronasal receptors and transcription factors.

not known. To explore our dataset for clues of transcriptional activities associated with VR expression, we plotted the cross-correlation between VRs and their TF profiles. We observed correlations among receptors (*Figure 4I and J*). We did not find an obvious association between TF profiles and VR sequence similarity of pairs (*Figure 4—figure supplement 2A*).

There is a high correlation among a large portion of V1Rs (*Figure 4I*). By analyzing the correlation between individual TFs with the VRs, we found that V1R expression is overwhelmingly associated with *Meis2* (*Figure 4—figure supplement 2B*). The few receptor types that do not show high correlation with *Meis2* are associated with *Egr1* or *Fos*. It is not clear whether these prototypical immediate early genes are involved in specifying receptor choice. This result indicates that once the V1R lineage is specified by *Meis2*, receptor choice is not determined by specific combinations of transcription factors. This scenario is similar to receptor choice by the OSNs in the MOE.

Different from the V1Rs, we observed islands of high similarity of TF expression among the V2Rs, indicating that receptors in these islands share the same set of TFs (*Figure 4J*). We identify correlation between individual TFs with V2Rs (*Figure 4—figure supplement 2C*). Whereas *Tfap2e* is involved in specifying the V2R fate, it is only associated with the expression of a subset of V2Rs. *Pou2f1* and *Atf5* are more strongly associated with other subsets of receptors. Unlike the V1Rs, there is a disparate set of TFs associated with the V2Rs, suggesting that the V2R choice may be determined by combinations of TFs. Based on the prevalence of individual transcription factors in association with the VRs, we propose a model of transcriptional cascade that may be involved in receptor choice (*Figure 4K*). For the V2Rs, *Tfap2e*+ cells can be further specified by *Ikzf4*, *Tcf4*, and *Trps1*. In *Ikzf4*-negative cells, *Batf3*, *Atf5*, *Pbx2*, and *Pou2f1* can specify receptor types, respectively. In the *Tfap2e*- cells, receptors can be specified by *Pou2f1*, *Rlf*, and *Batf3*.

## Co-expression of chemosensory receptors

We next investigated co-expression of receptors in individual cells. Since we applied SoupX to limit ambient RNA contamination and Scrublet to remove doublet cells, we set a stringent criterion in counting receptors expressed by single cells (*Young and Behjati, 2020*). V1R and V2R genes on average constitute ~2% of total reads per cell, and the ORs constitute less than 1% of the total reads per cell (*Figure 5—figure supplement 1A–C*). On average, the V2Rs have significantly more than one receptor per cell (*Figure 5—figure supplement 1B*). Using Shannon Index to measure uniqueness of receptor expression in individual cells, we found that the mature VSN and OSNs have relatively high specificity, but many cells show significantly higher index values, indicating that they expressed multiple receptors (*Figure 5A*). In support of this observation, there are significant representations of the second and third highest expressed receptors in all three types of neurons (*Figure 5—figure supplement 1D–G*).

To further reduce contributions of spurious, random low-level expression, we only consider those receptors with at least 10 raw counts per cell and are found in at least five cells to evaluate co-expression among receptors. We split cells into groups according to cell-type, age, and neuronal lineage and plot the percentage of cells expressing zero, one, two, and three or more species (*Figure 5B–C*). This analysis shows that receptor expression specificity increases as the lineages progressed from progenitors into mature neurons. Immature neurons have more cells co-expressing receptors than mature cells. More co-expressions are observed in the younger animals than the older ones. This line of evidence indicates that the co-expression we have found is not from experimental artifacts but reflects real biological events. Contamination would not be selectively enriched in immature cells and not present in the INP cells.

We performed Fisher's exact test using contingency tables for every pairing of expressed receptor genes. For the pairs that pass the test, we generated Circos plots to show the genomic loci for all significant V1R, V2R, and interclass pairs (*Figure 5D–F*). This analysis showed that 47.7% of co-expressed receptors are co-localized on the same chromosome.

We found a few sets of co-expressed VRs that would have strong implications for how pheromone signals are detected. *Vmn1r85* (*V1rj3*) and *Vmn1r86* are two receptors located next to each other on

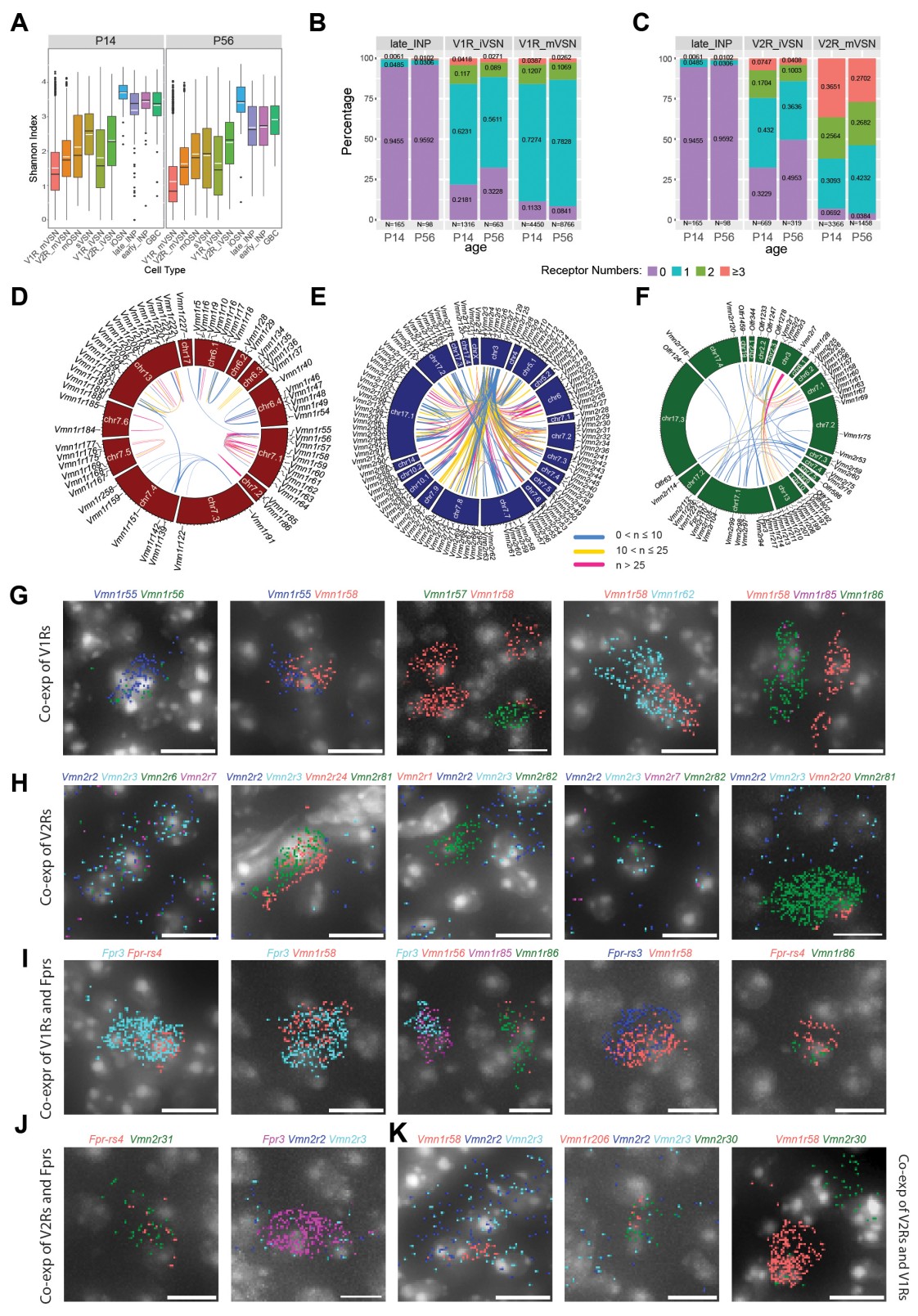

**Figure 5.** Co-expression among VNO receptor classes. (**A**) Shannon Indices showing the specificity of receptor expression. High values indicate more co-expressions. (**B–C**) Prevalence and level of receptor co-expression by age and cell- type for the V1R (**B**) and V2R (**C**) lineages, respectively. (**D–F**) Circos plot of genomic loci for significantly co-expressed receptor pairs in the V1R (**D**) V2R, (**E**) and across-type populations (**F**). (**G–K**) Detection of

*Figure 5 continued on next page*

*Figure 5 continued*

receptor gene co-expression using Molecular Cartography. Individual dots represent single molecules. Colors represent different receptor genes. DAPI stain is shown as gray. Scale bar: 10 μm.

The online version of this article includes the following figure supplement(s) for figure 5:

**Figure supplement 1.** Receptor expression statistics.

Chr. 7, sharing a high level of homology, and belonging to the V1rj clade. They have a high level of co-expression but are not co-expressed with *Vmn1r89* (*V1rj2*), located ~100 Kb away, even though both *Vmn1r85* and *Vmn1r89* receptors are activated by sulfated estrogen and carry information about the estrus status of mature female mice (*Haga-Yamanaka et al., 2015*; *Nodari et al., 2008*; *Haga-Yamanaka et al., 2014*; *Figure 5D*). Another set of receptors that recognize female-specific phero-mone cues are the V1re clade receptors. We found that *Vmn1r185* (*V1re12*), which recognizes female identity pheromones, was co-expressed with its close homolog *Vmn1r184* gene, which is about 350 kb away on Chr 7. They are not co-expressed with *Vmn1r69* (*V1re9*), which also recognizes female pher-omones, but is located 16 Mb apart on Chr. 7 (*Haga-Yamanaka et al., 2015*; *Haga-Yamanaka et al., 2014*; *Fu et al., 2015*; *Lee et al., 2019*; *Figure 5D*).

On Chr. 7 there are two other major clusters of V1R genes that show co-expression. One cluster includes *Vmn1r55-Vmn1r64*, 10 genes belonging to the *V1rd* clade and located within a 600 Kb region. Another one includes *Vmn1r167*, *Vmn1r168*, and *Vmn1r169*, which appear to be specifically paired with *Vmn1r175*, *Vmn1r177*, and *Vmn1r176*, respectively. These receptor pairs are arranged in a 300 Kb region with a head-head orientation. Outside of Chr. 7, several small clusters on Chr. 6 and one large cluster on Chr. 3 exhibit significant co-expression of V1Rs.

The V2R neurons coordinately express one common V2R and one specific receptor (*Akiyoshi et al., 2018*; *Ishii and Mombaerts, 2011*). Our co-expression analysis confirms the broad association of *Vmn2r1-7*, which are located on Chr. 3, with other receptors across various chromosomal locations (*Figure 5E*). In addition, we also identified co-expression patterns among the specifically expressed V2Rs. Among these receptor genes, intra-chromosomal co-expressions are observed for receptors residing on Chr. 7, Chr. 17, and Chr. 5. There is also inter-chromosomal co-expression between one locus on Chr. 17 with a cluster on Chr. 7 (*Figure 5E*).

Lastly, we have observed co-expression of receptors across different classes of receptors (*Figure 5F*). Notably, *Vmn2r1-3* are co-expressed with several *Vmn1r* receptors on Chr. 3. *Vmn2r7* is co-expressed with a group of Vmn1r genes on Chr. 13. The odorant receptor *Olfr344* is co-expressed with several V1R and V2R genes.

In past studies, VR expression was examined using in situ hybridization, immunostaining, or genetic labeling. The traditional histological methods were not sensitive enough to quantitatively measure signal strength. Moreover, pairwise double in situ is too laborious to capture co-expression of two or more receptors. To verify co-expression of VR genes by individual VSNs, we selected 30 VR genes based on scRNA-seq data and used Molecular Cartography to examine their expression patterns in situ.

Given the high incidents of co-expression between receptors in the *Vmn1r55-64* cluster on Chr. 7 (*Figure 5D*), we included five probes for this set of genes and confirmed colocalization of pairs in the VSNs (*Figure 5G*), Interestingly, we did not find cells expressing more than two receptor genes for this set. We also confirmed the co-expression between *Vmn1r85* and *Vmn1r86* as indicated by single cell data (*Figure 5D and G*).

We confirmed the co-expression among genes from the V2R genes (*Figure 5H*; *Silvotti et al., 2011*; *Francia et al., 2015*). Consistent with previous reports (*Silvotti et al., 2011*; *Silvotti et al., 2007*), *Vmn2r2*, *Vmn2r3*, *Vmn2r6*, and *Vmn2r7* are comingled in several cells, but *Vmn2r1* is not co-ex-pressed with these four broadly expressed V2Rs (*Figure 5H*). Outside of the *Vmn2r1-7* group, we find that *Vmn2r81* is co-expressed with *Vmn2r20* or *Vmn2r24* but without any of the *Vmn2r1-7* transcripts (*Figure 5H*). We detected more incidents of co-expression between *Fpr3* and *Fpr-rs4*, and between *Fpr3*, *Fpr-rs3*, *Fpr-rs4* with V1Rs than with V2Rs (*Figure 5I and J*). We also found co-expressions that are not predicted by the single cell analysis (*Figure 5K*). The discrepancy likely can be attributed to the relatively low-level expression of one of the receptor genes. This type of co-expression may not pass the strict criteria we set for the single cell analysis.

## A surface molecule code for individual receptor types

VSNs expressing a given receptor type project to the AOB to innervate glomeruli distributed in quasi-stereotypical positions (*Rodriguez et al., 1999*; *Belluscio et al., 1999*; *Haga et al., 2010*). The high number of glomeruli innervated by a given VSN type raises the question about mechanisms that specify the projection patterns and the connection between the VSNs and the mitral cells. For a genetically specified circuit that transmits pheromone and other information to trigger innate behavioral and endocrine responses, there must be molecules that instruct specific connections among neurons. Several studies have revealed the requirement of *Kirrel2*, *Kirrel3*, Neuropilin2 (*Nrp2*), *Epha5*, and *Robo/Slit* in vomeronasal axon targeting to the AOB (*Brignall et al., 2018*; *Prince et al., 2013*; *Knöll et al., 2001*; *Walz et al., 2002*; *Cloutier et al., 2002*; *Knöll et al., 2003*; *Prince et al., 2009*). However, how individual guidance molecules or their combinations specify connectivity of individual VSN types is completely unknown. Here, we leverage the unbiased dataset to identify surface molecules that may serve as code for circuit specification.

We identified 307 putative axon guidance (AG) molecules, including known cell surface molecules involved in transcellular interactions and some involved in modulating axon growth. Using this panel, we calculated pairwise similarity between two VR genes, and the similarity in their guidance molecule expression. Analysis of the relationship indicates the there is a general increase in guidance molecule similarity associated with VR similarity (*Figure 6A*). Consistent with this observation, when we plotted the similarity of surface molecule expression among cells expressing different receptors, we found islands of similarity among the receptor pairs (*Figure 6B*). We then conducted correlation analysis between the guidance molecules with the V1Rs (*Figure 6C*) and V2Rs (*Figure 6D*). Consistent with previous reporting, we found that *Kirrel2* was associated with nearly half of the V1Rs and *Kirrel3* was mainly associated with V2Rs that project to the caudal AOB (*Brignall et al., 2018*; *Prince et al., 2013*). *Robo2* is associated with nearly all V2Rs. We found that Teneurin (*Tenm2* and *Tenm4*) and protocadherin (*Pcdh9*, *Pcdh10*, and *Pcdh17*) genes were associated with specific receptors (*Lee et al., 2008*; *Alkelai et al., 2016*). *Epha5*, *Pdch10*, *Tenm2*, *Nrp2* are also strongly associated with V1Rs with partial overlap with each other. *Pcdh9*, *Tenm4*, *Cntn4*, *EphrinA3*, *Pchd17*, as well as *Kirrel2* and *Kirrel3* all show association with specific V2Rs. Numerous guidance molecules that are not broadly expressed are also associated with individual VRs.

Based on these associations, and supported by existing literature, we propose a model of the specification of separate groups of VRs. In this model, *Robo2* separates the rostral vs. caudal AOB. *Robo2* +V2 R neurons project to the caudal AOB, whereas *Robo2*- V1R cells project to the rostral AOB (*Knöll et al., 2003*; *Prince et al., 2009*; *Cloutier et al., 2004*). Among V1Rs, *Kirrel2* expression distinguishes between two groups of cells (*Brignall et al., 2018*; *Wang et al., 2021*). The *Kirrel2* +population can be further separated into *Epha5* +and *Epha5*- population (*Prince et al., 2013*; *Wang et al., 2021*). The *Epha5* +population can be separated further by *Pcdh10* expression. In the *Kirrel2*- cells, *Epha5*, *Pcdh10*, *Tenm4*, and *Tenm2* mark separate groups. *Ncam1*, *Epha5*, and *Cntn4* may contribute to specifying small sets of cells. For the V2Rs, *Pcdh9*, *Cntn4*, *Tenm4*, *Tenm2*, and *Pcdh17* have a decreasing range of expression, which may be used to specify increasingly refined connection. Notably, even though *Kirrel2* and *Pcdh10* are mostly detected in the V1Rs, they are also expressed by small sets of the V2Rs.

## Transcriptional regulation of receptor and axon guidance cues

The specification of a vomeronasal circuit needs to be tied to receptor expression. We next address whether a transcriptional code is associated with both VR and AG molecule expression. We calculated pairwise similarity among the receptors according to their expression of TF and AG genes (*Figure 7A*). The result shows that V1R and V2R are distinctively separated. Besides a small group of broadly expressed V2Rs, all VR types are unique in their gene expression. Fpr types are more similar in their expression profile with the V1Rs, but the OR types are intermingled with both VR types.

To further determine the TF/AG associations that are specific to individual receptor types, we applied stringent criteria to calculate the Jaccard Index for each pair of TF/AG (*Figure 7B*), which reflect the statistical probability of co-expression within the same cell. The analysis reveals unique combinations for every receptor type (*Figure 7C–G*). Even though the only TF associated with V1R expression is *Meis2*, other TFs are involved in specifying AG gene expression. For example, cells expressing V1Rs with high sequence homology and in chromosomal proximity share a similar set of

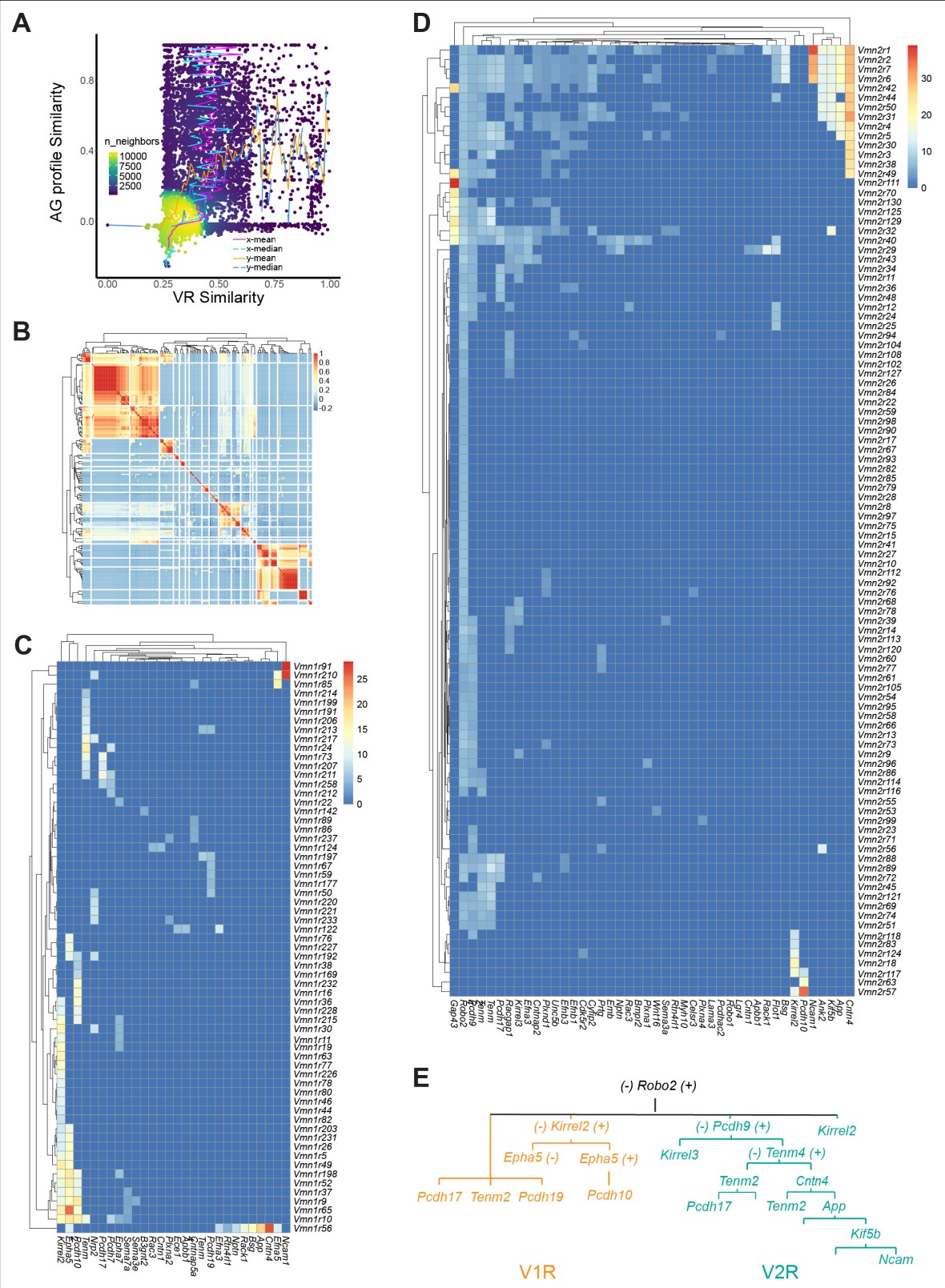

**Figure 6.** Axon guidance molecules associated with receptor genes. (**A**) Distribution density plot showing relationship between similarity of axon guidance (AG) gene expression profiles and receptor sequence similarity. The distribution of receptor similarity (x-mean and x-median) remains constant over the range of AG similarity. The AG similarity (y-mean and y-median) as a function of receptor similarity shows a strong correlation at the dense part of the curve. (**B**) Heatmap showing the Pearson correlation coefficient among VRs in their AG expression. (**C** and **D**) Heatmaps showing the V1R-AG

*Figure 6 continued on next page*

*Figure 6 continued*

(**C**) and V2R-AG (**D**) associations. Heat shows average expression level for AG genes for a given receptor type. (**E**) A simplified model of hierarchical distribution of AG in the mVSNs.

TF and AG genes, but the expression patterns are distinct from each other (*Figure 7D*). For broadly expressed V2Rs, *Vmn2r3* and *Vmn2r7*, which are co-expressed by the same set of cells, share nearly identical TF/AGs (*Figure 7E*). In contrast, *Vmn2r1*, which does not co-express with either *Vmn2r3* or *Vmn2r7*, lacks the expression of *Pou2f1* and *Tenm2* despite sharing all other genes. Other V2R types also show distinct TF/AG associations (*Figure 7F–G*).

## Discussion

Single cell RNA-seq analysis provides an unprecedented opportunity to identify cell types and determine genes associated with individual cells, but it is not without pitfalls. At the current state of the art, the depth of sequencing only allows sampling of transcripts that are expressed at relatively high levels. Sequencing dropouts and potential contamination also can complicate the analysis. Using the current state-of-the-art tools, and applying conservative criteria, we provide an in-depth look at the molecular and cellular organization of the mouse VNO. The analyses reveal new cell types, specific co-expression of receptors, and transcriptional regulation of lineage specification. Moreover, our analyses uncover specific associations between transcription factors, surface guidance molecules, and individual receptor types that may determine the wiring specificity in the vomeronasal circuitry.

The molecular distinction between the sVSNs and the classic VSNs indicates that they may serve a specific function. They are different from the solitary chemosensory cells that are trigeminal in nature (*Ogura et al., 2010*). We speculate that these cells may secrete olfactory binding proteins and mucins in response to VNO activation. The VNO is a semi-blind tubular structure. Chemical cues are actively transported into the VNO and can only be cleared by active transport systems, which are thought to be carried out by the lipocalin family of proteins, or by degradation (*Ogura et al., 2010*; *Meredith et al., 1980*; *Wysocki et al., 1985*; *Miyawaki et al., 1994*). These proteins are important to protect the integrity of neuroepithelia. They are generally produced by the SCs or the Bowman's gland (*Miyawaki et al., 1994*). It is plausible that the sVSNs can produce specific lipocalin proteins based on the ligand detected by the VRs they express. These cells may also convey chemosensory information to the AOB, but we do not know if they project to the central brain. We also have identified a class of canonical OSNs in the VNO. Previous reports show that these neurons project to AOB. It is plausible that the OSNs can detect a set of volatile odors that carry species-specific information and directly convey it to brain areas that regulate innate responses. Our list of these ORs could direct effort to identify these odors to reveal their ethological relevance.

The co-expression of multiple VRs in individual VSNs is intriguing, as a previous analysis of the MOE detected minimal co-expression of ORs (*Hanchate et al., 2015*). Importantly, there is a higher propensity for VR co-expression between certain receptor pairs. Notably, there is co-expression of receptors sharing common ligands, or that are similar in their sequences. These observations indicate that co-expressed receptors may serve redundant function detecting the cues. For example, the *V1rj* receptors are cognate receptors for sulfates estrogen and estrus signals. Their co-expression indicates that the neurons detect the same class of molecules redundantly. Moreover, co-expression of similarly tuned receptors makes it plausible for heterotypic convergence, when these neurons converge into glomeruli that express one or the other receptors.

How specificity of neuronal connections in the olfactory system is determined remains unknown. In the main olfactory system, glomerular positions are coarsely specified along the anterior-posterior and dorsal-ventral axes by gradients of axon guidance molecules whereas the sorting of axons according to the odorant receptors is mediated by homophilic attraction and heterotypic repulsion using a different set of guidance molecules (*Mori and Sakano, 2011*). Spontaneous neural activities determine the expression of both sets of molecules. It is not known whether VSNs utilize the same mechanism. Given the multi-glomerular innervation patterns by the VSNs, it is exceedingly difficult to determine the contribution of individual guidance molecules to specifying VSN innervation.

We have identified guidance molecules associated with individual VRs that potentially constitute a code set that specifies VSN axon projections and their connection with postsynaptic cells. Each

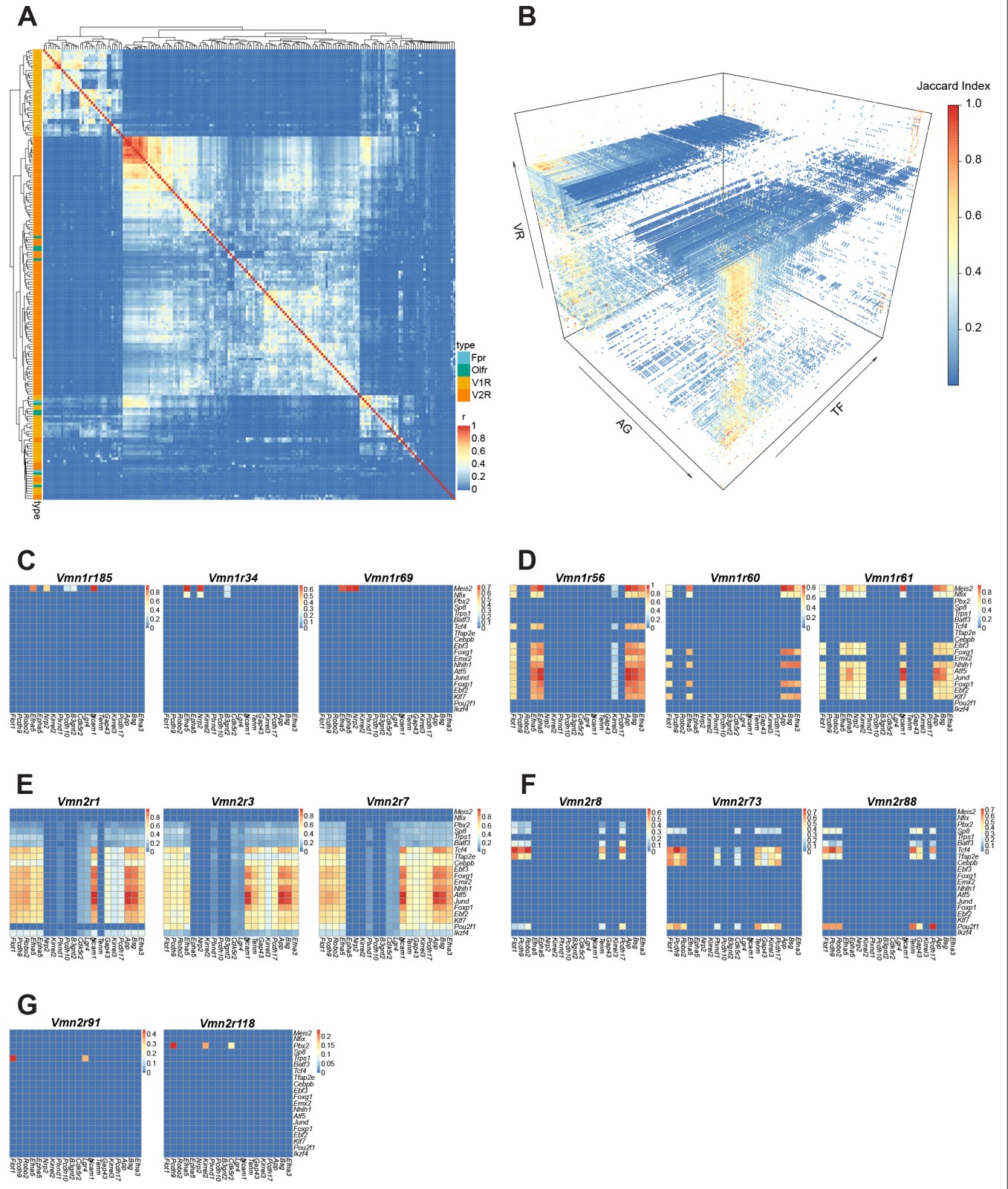

**Figure 7.** Transcriptional determinants of axon guidance molecules for individual receptor types. (**A**) Correlation heatmap between receptor types calculated from co-expressed AG and TF genes. Receptor types are color-coded. (**B**) 3-D heatmap showing the Jaccard Indices between AG and TF genes for each of the VRs in the dataset. (**C–G**) Heatmaps showing Jaccard Indices of TF-AG associations for V1R (**C and D**) and V2R types (**E–G**). The lists of TF and AG genes here are abridged from the full list to enhance visualization. (**C**) These receptors share Meis2 expression but different AG

*Figure 7 continued on next page*

*Figure 7 continued*

genes. Note that Vmn1r185 and vmn1r69 both recognize female identify pheromones. (**D**) Similarity and distinction of AG/TF expression for three V1R types that are located in the same genomic location and with high sequence homology. (**E and F**) Shared TFs and AG genes for broadly (**E**) and uniquely (**F**) expressed V2R types. (**G**), distinct TFs and AGGs for uniquely expressed V2R types.

receptor type has a unique combination of guidance molecules expressed, which provides a basis for axon segregation and convergence. There are a few molecules that are shared broadly by various VSN types. These can be used to instruct general spatial locations of the VSN axons. For example, *Robo2* separates the anterior vs. posterior AOB. Knockout of *Robo2* causes mistargeting of V2R neurons to the rostral AOB. Our models also indicate that *Kirrel2* and *Kirrel3* are expressed by nearly half of the VR types in partially overlapping patterns. Deletion of *Kirrel2* or *Kirrel3* leads to disorganization of glomeruli in the posterior AOB. Protocadherins and tenurins add new dimensions to this code. We also identified several guidance molecules that are more specifically associated with individual VRs. They could provide additional cues to separate axons that share broadly expressed guidance molecules.

We have identified lineage relationships among cells in the VNO and a dynamic transcriptional cascade that likely specifies cell types during development. While our model agrees with that of *Katreddi et al., 2022* on the main transcription factors that specify the lineage, it adds more details on both the induction and suppression of genes in specifying the cell fate. For example, we confirm *Meis2* and *Tfap2e* as transcription factors that maintain the V1R and V2R fate, but we also found that the down regulation of *Neurog1*, but not *Neurod1*, is associated with a transition from early INPs to late INPs. The downregulation of *Sp8*, *Nfib*, and *Bcl11b* is likely important for committing to the V1R lineage for late INPs. On the other hand, downregulation of *Sp8* and upregulation of *Fezf1*, *Olig2*, and *Tshz2* likely set up commitment to the OSN fate. We also find that in all three lineages, the expression of *Tshz2* is associated with transition to the immature neuronal fate from the late INPs.

We observed a striking difference between V1R and V2R VSNs in the transcription factors associated with receptor choice. There is no overt association between V1R with specific transcription factors. This observation is reminiscent of OSNs in the olfactory epithelium, where OR expression is stochastic and mediated by de-repression of epigenetically silenced OR loci (*Monahan et al., 2019*; *Dalton et al., 2013*; *Clowney et al., 2012*; *Lyons et al., 2013*; *Monahan and Lomvardas, 2015*; *Lomvardas et al., 2006*). The absence of specific transcription factors with individual V1R choice suggests that a similar mechanism may operate in the V1R VSNs. Monoallelic expression of *V1Rb2* supports this notion (*Rodriguez et al., 1999*). On the other hand, we observed that for individual V1R types, there are specific associations between transcription factors with guidance molecules. This observation implies that the expression of guidance molecules is determined by combinations of transcription factors even though these transcription factors may not determine V1R expression. This is also reminiscent of the OSNs, where the expression of guidance molecules is determined by spontaneous neural activities (*Imai et al., 2006*; *Nakashima et al., 2013*; *Serizawa et al., 2006*). That is, once the receptor choice is made, the specific receptor being expressed determines the guidance molecules to specify their projection patterns.

In direct contrast, V2R VSNs likely use combinations of transcription factors to specify receptor expression as well as guidance molecules. Some of the transcription factors that we observe to be associated with V2R expression are also associated with guidance molecule expression. For example, *Pou2f1*, *Atf5*, and *Zfp268* are involved in both processes.

## Materials and methods

**Key resources table**

| Reagent type (species) or resource | Designation | Source or reference | Identifiers | Additional information |
|---|---|---|---|---|
| Strain, strain background (*Mus musculus*; 2 females, 2 males) | C57BL/6 | In-house breeding | | |
| Strain, strain background (*Mus musculus*; 4 males, 4 females) | CD-1 | In-house breeding | | |

*Continued on next page*

*Continued*

| Reagent type (species) or resource | Designation | Source or reference | Identifiers | Additional information |
|---|---|---|---|---|
| Commercial assay or kit | Chromium Next GEM Single Cell 3' GEM, Library and Gel Bead Kit v3.1 | 10 X Genomics | 1000120 | |
| Commercial assay or kit | ChromiumSingle Cell 3' GEM, Library & Gel Bead Kit v3.0 | 10 X Genomics | 1000075 | |
| Commercial assay or kit | DNase I (RNase free) | NEB | M0303 | |
| Commercial assay or kit | Papain | Calbiochem | 5125 | |
| Commercial assay or kit | DAPI | Thermo Fisher Scientific | 62247 | |
| Commercial assay or kit | L-cysteine | Calbiochem | 243005 | |
| Commercial assay or kit | BSA | Sigma-Aldrich | A8806 | |
| Commercial assay or kit | Frozen section media | Leica | 3801481 | |
| Commercial assay or kit | DRAQ5 | Invitrogen | 65-0880-96 | |
| Commercial assay or kit | HBSS | VWR | VWRL0121-0500 | |
| Commercial assay or kit | PBS | Gibco | 10010023 | |
| Commercial assay or kit | Urethane | Sigma-Aldrich | U2500 | |
| Commercial assay or kit | NovaSeq S1 | Illumina | 20012865 | |
| software, algorithm | Fiji ImageJ | *Goldstein et al., 2018* | https://imagej.net/software/fiji/ | |
| Software, algorithm | QuPath v0.4.3 | *Bankhead et al., 2017* | https://qupath.github.io | |
| Software, algorithm | Seurat | *Hao et al., 2021* | https://satijalab.org/seurat/ | |
| Software, algorithm | kallisto \| bustools | *Melsted et al., 2019* | https://www.kallistobus.tools | |
| Software, algorithm | DropletUtils | *Lun et al., 2019* | https://bioconductor.org/packages/release/bioc/html/DropletUtils.html | |
| Software, algorithm | Illustrator | Adobe | https://www.adobe.com/illustrator | |
| Software, algorithm | SoupX | *Young and Behjati, 2020* | https://cran.r-project.org/web/packages/SoupX/index.html | |
| Software, algorithm | clustree | *Zappia and Oshlack, 2018* | https://cran.r-project.org/web/packages/clustree/index.html | |
| Software, algorithm | ggplot2 | *Wickham et al., 2016* | https://cran.r-project.org/web/packages/ggplot2/index.html | |
| Software, algorithm | glmGamPoi | *Ahlmann-Eltze and Huber, 2021* | https://bioconductor.org/packages/release/bioc/html/glmGamPoi.html | |
| Software, algorithm | vegan | *Oksanen et al., 2019* | https://cran.r-project.org/web/packages/vegan/index.html | |
| Software, algorithm | Scrublet v0.2.3 | *Wolock et al., 2019* | https://github.com/swolock/scrublet | |
| Software, algorithm | reticulate | *Ushey et al., 2017* | https://cran.r-project.org/web/packages/reticulate/index.html | |
| Software, algorithm | GeneOverlap | *Shen, 2019* | https://bioconductor.org/packages/release/bioc/html/GeneOverlap.html | |
| Software, algorithm | circlize | *Gu et al., 2014* | https://cran.r-project.org/web/packages/circlize/index.html | |

*Continued on next page*

*Continued*

| Reagent type (species) or resource | Designation | Source or reference | Identifiers | Additional information |
|---|---|---|---|---|
| Software, algorithm | Slingshot | *Street et al., 2018* | https://www.bioconductor.org/packages/release/bioc/html/slingshot.html | |
| Software, algorithm | tradeSeq | *Van den Berge et al., 2020* | https://www.bioconductor.org/packages/release/bioc/html/tradeSeq.html | |
| Software, algorithm | msigdbr | *Dolgalev, 2020* | https://cran.r-project.org/web/packages/msigdbr/vignettes/msigdbr-intro.html | |
| Software, algorithm | fgsea | *Korotkevich et al., 2021* | https://bioconductor.org/packages/release/bioc/html/fgsea.html | |
| Software, algorithm | Molecular Cartography | Resolve Biosciences | https://resolvebiosciences.com/ | |

## Resource availability

### Lead contact

Further information and requests for resources and reagents should be directed to and will be fulfilled by C. Ron Yu (https://www.ryu@stowers.org).

### Data and code availability

All RNA-seq data are available from the NCBI GEO server (GSE252365). All original data generated in this study will be available for download at Stowers original data repository upon publication. No custom generated computer code was used for analysis.

An HTML file containing relevant figures and statistics from the study, as well as tables showing co-expression and differential expression results, can be accessed at the following URL: https://ronyulab.github.io/VNO-Atlas/.

## Experimental model and subject details

Wildtype CD1 postnatal day 14 (P14) pups and adults (P56) were used for the experiment. Both sexes were randomly assigned to the experiment. All animals were maintained in Stowers LASF with a 14:10 light cycle and provided with food and water ad libitum. Experimental protocols were approved by the Institutional Animal Care and Use Committee (IACUC) (#2022–151) at Stowers Institute and in compliance with the NIH Guide for Care and Use of Animals.

## Methods

### scRNA library preparation and sequencing

Mice VNOs were dissected in cold oxygenated ACSF following *Ma et al., 2011*. Dissected epitheliums were dissociated in papain solution (20 mg/mL papain and 3 mg/mL L-cysteine in HBSS) with RNase-free DNase I (10unit) at 37 °C for 15–20 mins. 0.01% BSA in PBS was added to the digestion solution before filtering with 70 µm and 30 µm filters (pluriSelect).

Dissociated cells were washed twice in 0.01% BSA with final volume 1 mL, followed by Draq5 (25 µM) and DAPI (0.5 µg/mL) staining 5 min on ice. Draq5+/DAPI- cells (live/nucleated cells) were sorted on BD Influx cytometer (BD Bioscience) with 100 µm nozzle. Dissociated sorted cells were assessed for concentration and viability via Luna-FL cell counter (Logos Biosystems). Cells deemed to be at least 90% viable were loaded on a Chromium Single Cell Controller (10 x Genomics), based on live cell concentration. Libraries were prepared using the Chromium Next GEM Single Cell 3' Reagent Kits v3.1 (10X Genomics) according to manufacturer's directions. Resulting cDNA and short fragment libraries were checked for quality and quantity using a 2100 Bioanalyzer (Agilent Technologies) and Qubit Fluorometer (Thermo Fisher Scientific). With cells captured estimated at ~5500–8000 cells per sample, libraries were pooled and sequenced to a depth necessary to achieve at least 40,000 mean reads per cell on an Illumina NovaSeq 6000 instrument utilizing RTA and instrument software versions current at the time of processing with the following paired read lengths: 28*8*91 bp.

## scRNA-Seq pre-processing and QC-filtering

Gene-by-cell barcode count matrices were generated from raw FASTQ files using the kallisto|bustools (v0.48.0|v0.41.0; *Bray et al., 2016*; *Melsted et al., 2019*; *Melsted et al., 2021*) workflow with the *Mus musculus* genome assembly GRCm39 (mm39) and GTF gene annotation files retrieved from ENSEMBL release 104 (*Howe et al., 2021*). All downstream QC-filtering and analysis was performed in an R environment (v4.0.3) (*R Development Core Team, 2013*). Empty droplets were estimated and filtered from the data using the *barcodeRanks* function of the DropletUtils package (v1.10.3; *Regev, 2019*; *Lun et al., 2019*) with the lower bound of UMI-counts set to 100. The count data was then imported into R (v4.0.3) using the Seurat package (v4.3.0; *Hao et al., 2021*) and ambient RNA contamination was automatically estimated with the *autoEstCont* function and removed with the *adjustCounts* function from the SoupX package (v1.5.2; *Young and Behjati, 2020*). Cell barcodes representing multiplets were identified and removed with Scrublet (v0.2.3; *Wolock et al., 2019*) interfacing with Python using reticulate (v1.30; *Ushey et al., 2017*; *Kevin and Tang, 2023*). Cell barcodes expressing <750 genes, >2.5 standard deviations above the mean number of genes or counts, or >5% of reads originating from mitochondrial genes were removed from the downstream analysis.

## scRNA-Seq integration and clustering

To cluster cells from multiple samples, raw gene counts for each sample were normalized with the *SCTransform* function (v0.3.2; *Hafemeister and Satija, 2019*) using the *glmGamPoi* method from the glmGamPoi package (v1.6.0; *Ahlmann-Eltze and Huber, 2021*), samples were then integrated using the Seurat integration pipeline. After principal component analysis (PCA) the dimensionality of the whole integrated VNO dataset was estimated to be 25 based on visual identification of an 'elbow' using the output from the *ElbowPlot* function. The shared nearest neighbor graph was constructed with the *FindNeighbors* function. Then, using the *FindClusters* function, an optimal resolution of 0.7 was chosen for the discovery of broad cell types in the VNO by running the *clustree* function from the clustree package (v0.30; *Zappia and Oshlack, 2018*) to evaluate cluster-level mean expression for a curated list of VNO cell-type marker-genes in a range of resolutions between 0 and 1, with intervals of 0.1; optimal resolution was considered the lowest resolution at which the expression of Neurod1 and Ascl1 diverge into two clusters, as these genes are unique marker-genes for immediate neural progenitors (INPs) and GBCs respectively. At 0.7 resolution 31 clusters were observed.

## Differential gene expression analysis and cell labeling

Clusters were assigned to cell-types using a list of marker-genes: Omp was used to broadly mark mature neurons, Gnai2, Gng13, and Meis2 marked the V1R subtype, Gnao1, Robo2, and Tfap2e marked the V2R subtype, Gnal and Cnga2 marked OSNs, Gap43, Stmn2, Bcl11b, and Lhx2 marked immature neurons, Neurod1 and Neurog1 marked INPs, Ascl1 and Ccnd1 marked GBCs, Krt5 and Krt14 marked HBCs, Sox9 and Hepacam2 marked MVs, Fezf2 and Sox2 marked SCs, S100b and Plp1 marked OECs, Cd34 and Cdh5 marked Endothelial cells, Acta2, Col1a2, and Mgp marked LP cells, Dock2 marked immune cells, Cx3cr1 and Ctss marked Microglia, and Il7r and Trbc2 marked T-cells. First, we ran the FindAllMarkers function on the clustered normalized counts, limited to genes present in ≥50% of each cluster's cells with an absolute value log2 fold-change ≥0.5; additionally, we ran the FeaturePlot function to plot expression of the marker-genes in 2D UMAP space. Clusters were then manually assigned to one cell-type both by visually ascertaining marker-gene expression overlap with cluster identity and by statistically validating significant marker-gene enrichment within cluster.

## Neuronal lineage

Cells labeled as GBC, INP, iVSN, iOSN, mVSN, mOSN, or sVSN were subset from the integrated whole VNO dataset, split by animal sample, then re-integrated and re-clustered, as described above. The *FindNeighbors* function was run using 12 PCs and the *FindClusters* function was run with a resolution of 6.0 which resulted in 84 clusters. Clusters were assigned to cell-types as described previously, while a further distinction between early and late INPs was inferred from the expression of Ascl1, Neurod1, Neurog1, and Gap43 in 2D UMAP space.

Differential gene expression analysis was performed between the novel sVSN cluster and the mature V1R, V2R, and OSN clusters, independently, running the *FindMarkers* function with the *logfc.*

*threshold* and *min.pct* parameters set to 0 to accommodate downstream gene set enrichment analysis (GSEA).

## Gene set enrichment analysis

Gene ontology (GO) terms from the biological process, molecular function, and cellular compartment categories along with their associated gene sets were retrieved with the *msigdbr* function from the msigdbr package (v7.5.1; *Dolgalev, 2020*). Ranked Wald test differential expression results between sVSNs and V1R/V2R mVSNs, respectively, were input into the *stats* parameter of the *fgsea* function from the fgsea package (v1.20.0; *Korotkevich et al., 2021*; *Subramanian et al., 2005*) along with the GO term gene sets. The top significant GO terms were plotted using the *ggplot* function from the ggplot2 package.

## Sex and age differences

To examine broad differences in gene expression between male and female mice and between P14 and P56 mice, we ran the FindMarkers function on the normalized count data using all cells in the neuronal lineage, all genes present in the data, and no threshold on log2 fold-change.

Significant differential gene expression results (padj ≤0.05) from the male/female and the P14/P56 test were used for GSEA, and the results were plotted using the ggplot function from the ggplot2 package.

## Immediate neural progenitors and immature vomeronasal sensory neurons

Cells previously identified as early and late INP, iVSN, iOSN, or mOSN, were subset from the neuronal dataset, split and reintegrated, as above, using 15 PCs and a resolution of 2.5, resulting in 30 clusters. Differential gene expression analysis was performed with the *FindMarkers* function between two clusters showing either V1R or V2R like properties but previously identified uniformly as early INPs in the whole neuronal lineage dataset. The *FeaturePlot* function, from the Seurat package, was used to show normalized expressions for genes of interest.

## Trajectory inference and differential gene expression analysis

### V1R/V2R lineage determination

To explore transcriptional differences over pseudotime between V1R and V2R VSNs, we subset GBCs, early and late INPs, V1R and V2R iVSNs, and mVSNs from the neuronal dataset, split the data by sample and reintegrated, then performed PCA. Trajectory inference analysis was performed with the *Slingshot* function from the Slingshot package (v1.8.0; *Street et al., 2018*) using the first 12 PCs and cell type labels from the neuronal dataset, with an input starting cluster of GBCs and two input end clusters for V1R and V2R mVSNs. To determine the *nknots* parameter for the *fitGAM* function from the tradeSeq package (v1.4.0; *Van den Berge et al., 2020*), we ran the *evaluateK* function with the raw count matrix, and the pseudotime and cell weight values output from *Slingshot*. We then ran the *fitGAM* with nknots = 5. We then ran the *patternTest* function to test for differences in gene expression patterns over pseudotime between the V1R and V2R lineages.

### Pseudotime analysis of sVSNs

To determine whether sVSNs represent an immature version of canonical VSNs, we performed trajectory inference analysis on the neuronal lineage with only cells belonging to the OSN lineage removed. Using the *Slingshot* function, we input the first 5 PCs and set the starting cluster to GBCs and three end-clusters to V1R mVSNs, V2R mVSNs, and sVSNs, respectively. Pseudotime values were assigned to cells based on their lineage membership and were subsequently plotted with the FeaturePlot function.

## Gene co-expression analysis

### VR co-expression

To investigate cell-level diversity of VR species across all cells in the neuronal dataset we calculated the Shannon diversity index for the raw gene counts for all Vmn1r, Vmn2r, Olfr, and Fpr genes using the *diversity* function from the vegan R package (v2.6–4) (*Oksanen et al., 2019*) with default parameters.

To determine what proportion of cells in the neuronal lineage had one, two, or three or more receptor species, we set a threshold of ≥10 raw counts for a receptor to be considered 'present'. To test whether co-expressing VR species were significant, using the same raw-count threshold of ≥10, we gathered a list of all cell barcodes where a receptor was observed, for all receptors. Using the lists of cell barcodes associated with the receptors, we ran the *newGOM* function from the GeneOverlap R package (v1.26.0; *Shen, 2019*), which calculates p-values using Fisher's exact test on a contingency table. p-Values were then corrected for multiple testing using the Benjamini-Hochberg procedure.

Using the circlize R package (v0.4.15; *Gu et al., 2014*), we plotted all co-expressed receptor pairs on circos plots showing each receptors genetic location and the number of cells expressing the pair, for all significant pairings (padj ≤0.05).

## VR co-expression with axon guidance (AG) and transcription factor (TF) genes

To ascertain VR co-expression with AG genes expressed at the plasma membrane, and with DNA binding TF genes, respectively, we used the Mouse Genome Informatics database to find all genes associated with the biological process gene ontology (GO) term 'axon guidance', or with the molecular function GO term 'DNA-binding transcription factor activity'. For the axon guidance gene set, we subset genes that were expressed at the plasma membrane. We set the VR raw count threshold to ≥10 counts and the AG and TF gene raw count threshold to ≥3. Then, using the cell barcodes associated with each VR and the cell barcodes associated with candidate AG and TF genes, we ran the *newGOM* function to find significant co-expression (padj ≤0.05) for all VRxAG and VRxTF pairs.

## VR-specific co-expression of AG and TF genes

To test whether AGs and TFs co-expressed for a given VR, we gathered all cell-barcodes where there was significant VRxAG or VRxTF co- expression. Then, using the same contingency table scheme as above, we looked for significant co-expression (padj ≤0.05) between all AGs and TFs previously found to co-express with a given VR.

## Spatial transcriptomics

### Samples

VNO tissues were dissected from 7 to 8 weeks old C57BL/6 J mice. Briefly, mice were anesthetized with urethane at a dose of 2000 mg/kg body weight. Following general anesthesia, mice heads were decapitated, and the lower jaw was removed by cutting the mandible bone with scissors. The ridged upper palate tissue was peeled off to expose the nasal cavity. A surgical blade was inserted between the two upper incisors to expose the VNO. The whole VNO was carefully extracted by holding onto the tail bone and slowly lifting it up from the nasal cavity.

The dissected VNO was immediately transferred to cold 1 X PBS on ice, and subsequently embedded in frozen section media (Leica Surgipath FSC 22, Ref # 3801481) and frozen on liquid nitrogen. Frozen samples were sectioned at 10 μm thickness using the Thermo Scientific CryoStar NX70 cryostat. VNO sections were placed within capture area of cold slides that were provided by Resolve Biosciences. Slides were sent to Resolve Biosciences on dry ice for spatial transcriptomics analysis. Resolve Molecular Cartography protocols remain proprietary and were not disclosed. The probe design, tissue processing, imaging, spot segmentation, and image preprocessing were all performed using the Resolve Biosciences platform. Names and ENSEMBL IDs for genes probed are available in this study's public repository at https://ronyulab.github.io/VNO-Atlas/.

### Analysis

Regions of interest in the VNO were selected on brightfield images provided by Resolve Biosciences using Fiji ImageJ (*Goldstein et al., 2018*). Cell segmentation on final images was performed in QuPath v0.4.3 (*Bankhead et al., 2017*). Detected gene transcripts were then assigned to the segmented cells, thereby creating a gene-count matrix for each sample. To predict cell types, count matrices for all samples were imported into R then normalized using the *SCTransform* function with the *glmGamPoi* method. The samples were then integrated using the Seurat integration pipeline. Both the integrated whole VNO scRNA-seq dataset and the integrated Resolve molecular cartography dataset were

renormalized with *SCTransform* with the default method using ncells = 3000, then *RunPCA* was called on the renormalized data. Using the whole VNO scRNA-seq dataset as a reference and the molecular cartography dataset as a query, we ran the *FindTransferAnchors* function, then we ran the *TransferData* function to create a table of prediction score values for each cell in the spatial dataset. Cells were then labeled by type using the maximum prediction score for each cell.

Images showing co-localization of VRs were obtained in ImageJ using genexyz Polylux (v1.9.0) tool plugin from Resolve Biosciences.

## Region of neurogenesis

To test the hypothesis that neurogenesis occurs in the marginal zone of the VNO we first set a minimum cell-type prediction-score threshold of ≥ 0.3; all cells below the threshold were labeled 'unknown'. Then we used the simple features R package, sf (v1.0.16) to delineate regions of interest in the VNO. We excluded the non-neuronal region of the VNO from the analysis. Using the intersectional boundary between neural and non-neuronal epithelia as the center, we quantified the cells falling within a 750-pixel radius as in the marginal zones. Those fall out of the 750-pixel but within a 1500-pixel radius were quantified as in the intermediate zones. All remaining cells were classified as occurring in the 'main zone'. Plots displaying the results were created using the function, *ggplot,* from the R package ggplot2 (*Wickham et al., 2016*).

## Acknowledgements

We thank McKenzie Treese, KyeongMin Bae, Fang Liu, and members of Lab Animal Support Facility at Stowers for their technical support. We would like to acknowledge the University of Kansas Medical Center's Genomics Core for their support in generating data on the Illumina NovaSeq 6000 System. The core is supported by the following grants: Kansas Intellectual and Developmental Disabilities Research Center (NIH U54 HD 090216), the Molecular Regulation of Cell Development and Differentiation – COBRE (P30 GM122731-03) and the NIH S10 High- End Instrumentation Grant (NIH S10OD021743). This work was supported by fundings from NIH (R01DC008003 and R01DC020368) and Stowers Institute for Medical Research to CRY.

## Additional information

### Funding

| Funder | Grant reference number | Author |
| --- | --- | --- |
| National Institute on Deafness and Other Communication Disorders | R01 DC008003 | C Ron Yu |
| National Institute on Deafness and Other Communication Disorders | R01 DC020368 | C Ron Yu |

The funders had no role in study design, data collection and interpretation, or the decision to submit the work for publication.

### Author contributions

Max Henry Hills, Data curation, Software, Formal analysis, Validation, Investigation, Visualization, Methodology, Writing – review and editing; Limei Ma, Conceptualization, Investigation; Ai Fang, Seth Malloy, Investigation; Thelma Chiremba, Formal analysis, Investigation, Visualization; Allison R Scott, Anoja G Perera, Resources; C Ron Yu, Conceptualization, Supervision, Funding acquisition, Methodology, Writing - original draft, Project administration

### Author ORCIDs

Max Henry Hills ⓘ https://orcid.org/0009-0004-8464-9989
C Ron Yu ⓘ https://orcid.org/0000-0003-1555-8683

## Ethics

Experimental protocols were approved by the Institutional Animal Care and Use Committee (IACUC) (#2022-151) at Stowers Institute and in compliance with the NIH Guide for Care and Use of Animals.

Reviewer #1 (Public review): https://doi.org/10.7554/eLife.97356.3.sa1
Reviewer #2 (Public review): https://doi.org/10.7554/eLife.97356.3.sa2
Reviewer #3 (Public review): https://doi.org/10.7554/eLife.97356.3.sa3
Author response https://doi.org/10.7554/eLife.97356.3.sa4

# Additional files

## Supplementary files

• MDAR checklist

## Data availability

Sequencing data have been deposited in GEO under accession codes GSE252365.

The following previously published dataset was used:

| Author(s) | Year | Dataset title | Dataset URL | Database and Identifier |
|---|---|---|---|---|
| Hills M, Ma L, Fang A, Chiremba T, Malloy S, Scott A, Perera A, Yu C | 2024 | Molecular, Cellular, and Developmental Organization of the Mouse Vomeronasal organ at Single Cell Resolution | https://www.ncbi.nlm.nih.gov/geo/query/acc.cgi?acc=GSE252365 | NCBI Gene Expression Omnibus, GSE252365 |

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
