## [Editor Report · eLife Assessment]

The manuscript by Hills, et al. presents data that support multiple conclusions regarding the gene expression patterns of cells, especially chemosensory neurons. The evidence is largely **solid**, with transcriptomic analysis combined and validated by spatially resolved expression in tissue sections, but is incomplete in other ways with some claims not fully supported. This large-scale single-cell transcriptomics dataset is an **important** resource, alongside a thorough exploration of the molecular features of the different cell types within the mouse vomeronasal organ, including expression of chemosensory receptors.

---

## [Referee Report · Reviewer #1 (Public review)]

Summary:

The authors comprehensively present data from single cell RNA sequencing and spatial transcriptomics experiments of the juvenile male and female mouse vomeronasal organ, with a particular emphasis on the neuronal populations found in this sensory tissue. The use of these two methods effectively maps the locations of relevant cell types in the vomeronasal organ at a level of depth beyond what is currently known. Targeted analysis of the neurons in the vomeronasal organ produced several important findings, notably the common co-expression of multiple vomeronasal type 1 receptors (V1Rs), vomeronasal type 2 receptors (V2Rs), and both V1R+V2Rs by individual neurons, as well as the presence of a small but noteworthy population of neurons expressing olfactory receptors (ORs) and associated signal transduction molecules. Additionally, the authors identify transcriptional patterns associated with neuronal development/maturation, producing lists of genes that can be used and/or further investigated by the field. Finally, the authors report the presence of coordinated combinatorial expression of transcription factors and axon guidance molecules associated with multiple neuronal types, providing the framework for future studies aimed at understanding how these patterns relate to the complex glomerular organization in the accessory olfactory bulb. Several of these conclusions have been reached by previous studies, partially limiting the overall impact of the current work. However, when combined, these results provide important insights into the cellular diversity in the vomeronasal organ that are likely to support multiple future studies of the vomeronasal system.

Strengths:

The comprehensive analysis of the data provides a wealth of information for future research into vomeronasal organ function. The targeted analysis of neuronal gene transcription demonstrates the co-expression of multiple receptors by individual neurons, and confirms the presence of a population of OR-expressing neurons in the vomeronasal organ. Although many of these findings have been noted by others, the depth of analysis here validates and extends prior findings in an effective manner. The use of spatial transcriptomics to identify the locations of specific cell types is especially useful and produces a template for the field's continued research into the various cell types present in this complex sensory tissue. Overall, the manuscript's biggest strength is found in the richness of the data presented, which will not only support future work in the broader field of vomeronasal system function but also provide insights into others studying complex sensory tissues.

Weaknesses:

The inherent weaknesses of single cell RNA sequencing studies based on the 10x Genomics platforms (need to dissociate tissues, limited depth of sequencing, etc.) is acknowledged. However, the authors document their extensive attempts to avoid making false positive conclusions through the use of software tools designed for this purpose. Because of its complexity, there are some portions of the manuscript where the data are difficult to interpret as presented, but this is a relatively minor weakness. The data resulting from the use of the Resolve Biosciences spatial transcriptomics platform are somewhat difficult to interpret because the methods are proprietary and presented in an opaque manner. That said, the resulting data provide useful links between transcriptional identities and cellular locations, which is not possible without the use of such tools.

---

## [Referee Report · Reviewer #2 (Public review)]

In their paper entitled "Molecular, Cellular, and Developmental Organization of the Mouse Vomeronasal Organ at Single Cell Resolution" Hills Jr. et al. perform single-cell transcriptomic profiling and analyze tissue distribution of a large number of transcripts in the mouse vomeronasal organ (VNO). The use of these complementary tools provides a robust approach to investigating many aspects of vomeronasal sensory neuron (VSN) biology based on transcriptomics. Harnessing the power of these techniques, the authors present the discovery of previously unidentified sensory neuron types in the mouse VNO. Furthermore, they report co-expression of chemosensory receptors from different clades on individual neurons, including the co-expression of VR and OR. Finally, they evaluated the correlation between transcription factor expression and putative surface axon guidance molecules during the development of different neuronal lineages. Based on such correlation analysis, authors further propose a putative cascade of events that could give rise to different neuronal lineages and morphological organization.

We appreciate the authors' efforts to add context and citations that relate to recent single cell RNA sequencing studies in the VNO as well as to studies on vomeronasal receptors co-expression and V1R/V2R lineage determination. We also appreciate the new details on the marker genes used for cell annotation as well as clarifications about the differences between juvenile versus adult or male versus female samples.

A concern still remaining is that two major claims/interpretations - i.e., identification of canonical OSNs and a novel type sVSNs in the mouse VNO - either require experimental substantiation or the authors' claims should be toned down. In their response, Hills Jr. et al. acknowledge that their "paper is primarily intended as a resource paper to provide access to a large-scale single-cell RNA-sequenced dataset and discoveries based on the transcriptomic data that can support and inspire ongoing and future experiments in the field." The authors also write that given "the limited number of genes that we can probe using Molecular Cartography, the number of genes associated with sVSNs may be present in the non-sensory epithelium. This could lead to the identification of cells that may or may not be identical to the sVSNs in the non-neuronal epithelium. Indeed, further studies will need to be conducted to determine the specificity of these cells." Moreover, Hills Jr. et al. acknowledge that as "any transcriptomic study will only be correlative, additional studies will be needed to unequivocally determine the mechanistic link between the transcription factors with receptor choice. Our model provides a basis for these studies." We agree with all these points. Importantly, in the revised manuscript, the authors do not acknowledge that their primary intention is to present "a resource paper to provide access to a large-scale single-cell RNA-sequenced dataset", nor do they acknowledge any of the other caveats/limitations mentioned above. We believe that the authors should not only mention these aspects in their response to the reviews, but they should also make these intentions/caveats/limitations very clear in the manuscript text.

---

## [Referee Report · Reviewer #3 (Public review)]

This study presents a detailed examination of the molecular and cellular organization of the mouse VNO, unveiling new cell types, receptor co-expression patterns, lineage specification regulation, and potential associations between transcription factors, guidance molecules, and receptor types crucial for vomeronasal circuitry wiring specificity. The study identifies a novel type of VSN molecularly different from classic VSNs, which may serve as accessory to other VSNs by secreting olfactory binding proteins and mucins in response to VNO activation. They also describe a previously undetected co-expression of multiple VRs in individual VSNs, providing an interesting view to the ongoing discussion on how receptor choice occurs in VSNs, either stochastic or deterministic. Finally, the study correlates the expression of axon guidance molecules associated with individual VRs, providing a putative molecular mechanism that specifies VSN axon projections and their connection with postsynaptic cells in the accessory olfactory bulb.

The conclusions of this paper are well supported by data, but some aspects of data analysis and acquisition need to be clarified and extended.

(1) The authors claim that they have identified two new classes of sensory neurons, one being a class of canonical olfactory sensory neurons (OSNs) within the VNO. This classification as canonical OSNs is based on expression data of neurons lacking the V1R or V2R markers but instead expressing ORs and signal transduction molecules, such as Gnal and Cnga2. Since OR-expressing neurons in the VNO have been previously described in many studies, it remains unclear to me why these OR-expressing cells are considered here a "new class of OSNs." Moreover, morphological features, including the presence of cilia, and functional data demonstrating the recognition of chemosignals by these neurons, are still lacking to classify these cells as OSNs akin to those present in the MOE. While these cells do express canonical markers of OSNs, they also appear to express other VSN-typical markers, such as Gnao1 and Gnai2 (Fig 2B), which are less commonly expressed by OSNs in the MOE. Therefore, it would be more precise to characterize this population as atypical VSNs that express ORs, rather than canonical OSNs.

(2) The second new class of sensory neurons identified corresponds to a group of VSNs expressing prototypical VSN markers (including V1Rs, V2Rs, and ORs), but exhibiting lower ribosomal gene expression. Clustering analysis reveals that this cell group is relatively isolated from V1R- and V2R-expressing clusters, particularly those comprising immature VSNs. The question then arises: where do these cells originate? Considering their fewer overall genes and lower total counts compared to mature VSNs, I wonder if these cells might represent regular VSNs in a later developmental stage, i.e., senescent VSNs. While the secretory cell hypothesis is compelling and supported by solid data, it could also align with a late developmental stage scenario. Further data supporting or excluding these hypotheses would aid in understanding the nature of this new cell cluster, with a comparison between juvenile and adult subjects appearing particularly relevant in this context.

(3) The authors' decision not to segregate the samples according to sex is understandable, especially considering previous bulk transcriptomic and functional studies supporting this approach. However, many of the highly expressed VR genes identified have been implicated in detecting sex-specific pheromones and triggering dimorphic behavior. It would be intriguing to investigate whether this lack of sex differences in VR expression persists at the single-cell level. Regardless of the outcome, understanding the presence or absence of major dimorphic changes would hold broad interest in the chemosensory field, offering insights into the regulation of dimorphic pheromone-induced behavior. Additionally, it could provide further support for proposed mechanisms of VR receptor choice in VSNs.

(4) The expression analysis of VRs and ORs seems to have been restricted to the cell clusters associated to the neuronal lineage. Are VRs/ORs expressed in other cell types, i.e. sustentacular, HBC or other cells?

Review update:

I believe the novel discovery of two classes of sensory neurons within the VNO-canonical olfactory sensory neurons (OSNs) and secretory vomeronasal sensory neurons (sVSNs)-should be interpreted with caution. Firstly, these cell types are relatively rare, constituting less than 2% of total cells and only 2-6% of the neuronal population (according to Fig. S3). While the OSNs exhibit gene expression profiles consistent with canonical olfactory signal transduction and cilia-related gene ontology, key aspects such as their cell morphology (including the presence of cilia) and functional evidence for chemosignal detection have yet to be demonstrated. The neuronal lineage of sVSNs remains unclear to me. It is uncertain what developmental trajectories these cells follow: do they arise as a specialized subtype of V1R or V2R lineages, or do they have an independent lineage determination, similar to OSNs? At what stage does the commitment to the sVSN lineage begin-during the INP stage or the immature sensory neuron stage? A pseudotime inference analysis of sVSNs could help clarify these questions.

---

## [Author Response]

The following is the authors’ response to the original reviews.

**Reviewer #1:**
…several previous studies have identified co-expression of vomeronasal receptors by vomeronasal sensory neurons, and the expression of non-vomeronasal receptors, and this was not adequately addressed in the manuscript as presented.

We’ve added context and citations to the Introduction and Results sections relating to recent studies on the co-expression of vomeronasal receptors and the expression of non-vomeronasal receptors in VSNs.

The data resulting from the use of the Resolve Biosciences spatial transcriptomics platform are somewhat difficult to interpret, and the methods are somewhat opaque.

The Molecular Cartography platform relies on multi-plex imaging of fluorescent probes that bind specifically to individual gene transcripts to determine their spatial location. Unfortunately, the detailed protocols remain proprietary at Resolve Biosciences and were not disclosed. We have clarified this in the revised manuscript. Our role in the acquisition and processing of data for this experiment is included in the current Methods section. Additional analysis produced from the Molecular Cartography data have been added (See response to Reviewer #2, below) to the supplemental materials to help clarify interpretation of the results.

**Reviewer #2:**
…the authors present a biased report of previously published work, largely including only those results that do not overlap with their own findings, but ignoring results that would question the novelty of the data presented here.

We had no intention of misleading the readers. In fact, we have discussed discrepancies between our results with other studies. However, we inadvertently left out a critical publication in preparing the manuscript. We have added context and citations relating to recent studies that use single cell RNA sequencing in the vomeronasal organ, studies relating to the co-expression of vomeronasal receptors, and studies discussing V1R/V2R lineage determination. In Discussion, we also compared our model with a previous one of genetic determination of VNO neuronal fate.

Did the authors perform any cell selectivity, or any directed dissection, to obtain mainly neuronal cells? Previous studies reported a greater proportion of non-neuronal cells. For example, while Katreddi and co-workers (ref 89) found that the most populated clusters are identified as basal cells, macrophages, pericytes, and vascular smooth muscle, Hills Jr. et al. in this work did not report such types of cells. Did the authors check for the expression of marker genes listed in Ref 89 for such cell types?

For VNO dissections, we removed bones and blood vessels from VNO tissue and only kept the sensory epithelium. This procedure removed vascular smooth muscle cells, pericytes, and other non-neuronal cell types, which explains differences in cell proportions between our study and previous studies. We used a DAPI/Draq5 assay to sort live/nucleated cells for sequencing and no specific markers were used for cell selection. All cells in the experiment were successfully annotated using the cell-type markers shown in Fig. 1B, save for cells from the sVSN cluster, which were novel, and required further analysis to characterize.

The authors should report the marker genes used for cell annotation.

Marker genes used for cell annotation are shown in figure 1B. A full list of all marker genes used in the cell annotation process has been added to the Methods section.

The authors reported no differences between juvenile and adult samples, and between male and female samples. It is not clear how they evaluate statistically significant differences, which statistical test was used, or what parameters were evaluated.

The claims made about male/female mice and P14/P56 mice directly pertain to the distribution of clusters and cells in UMAP space as seen in Figure 1 C & D. We have performed differential gene expression analysis for male/female and P14/P56 comparisons using the FindMarkers function from the Seurat R package. Although we have found significant differential expression between male and female, and between P14 and P56 animals, the genes in this list do not appear to be influential for the neuronal lineage and cell type specification or related to cell adhesion molecules, which are the main focuses of this study. Nevertheless, we have added these results to the supplemental materials.

‘Based on our transcriptomic analysis, we conclude that neurogenic activity is restricted to the marginal zone.’ This conclusion is quite a strong statement, given that this study was not directed to carefully study neurogenesis distribution, and when neurogenesis in the basal zone has been proposed by other works, as stated by the authors.

We have used fourteen slides from whole VNO sections in our Molecular Cartography analysis to quantify the number of GBCs, INPs, and iVSNs predicted in the marginal zone, the intermediate zone, and main/medial zone. We have performed a Wilcoxon signed-rank test to check for the significant presence of GBCs, INPs, and iVSNs in the marginal zone over their presence in the main/medial zone. The results are included in new Figure S3. The result from this analysis justifies our claim that neurogenesis is restricted to the MZ. This claim is also supported by the 2021 study by Katreddi & Forni.

The authors report at least two new types of sensory neurons in the mouse VNO, a finding of huge importance that could have a substantial impact on the field of sensory physiology. However, the evidence for such new cell types is based solely on this transcriptomic dataset and, as such, is quite weak, since many crucial morphological and physiological aspects would be missing to clearly identify them as novel cell types. As stated before, many control and confirmatory experiments, and a careful evaluation of the results presented in this work must be performed to confirm such a novel and interesting discovery. The reported "novel classes of sensory neurons" in this work could represent previously undescribed types of sensory neurons, but also previously reported cells (see below) or simply possible single-cell sequencing artefacts.

The reviewer is correct that detailed morphological and physiological studies are needed to further understand these cells. This is an opinion we share. Our paper is primarily intended as a resource paper to provide access to a large-scale single-cell RNA-sequenced dataset and discoveries based on the transcriptomic data that can support and inspire ongoing and future experiments in the field. Nonetheless, we are confident that neither of the novel cell clusters are the result of sequencing artefacts. We performed a robust quality-control protocol, including count correction for ambient RNA with the R package, SoupX, multiplet cell detection and removal with the Python module, Scrublet, and a strict 5% mitochondrial gene expression cut-off. Furthermore, the cell clusters in question show no signs of being the result of sequencing artefacts, as they are physically connected in a reasonable orientation to the rest of the neuronal lineage in modular clusters in 2D and 3D UMAP space. The OSN and sVSN cell clusters each show distinct and self-consistent expressions of genes (new Figure S4H). Gene ontology (GO) analysis reveals significant GO term enrichment for both the sVSN (Fig. 2G) and mOSN clusters when compared to mature V1R and V2R VSNs, indicating functional differences. We have performed pseudotime analysis of sVSNs, differential gene expression and gene ontology analysis of mOSNs. The results are shown in the new Figure S6.

The authors report the co-expression of V2R and Gnai2 transcripts based on sequencing data. That could dramatically change classical classifications of basal and apical VSNs. However, did the authors find support for this co-expression in spatial molecular imaging experiments?

Genes with extremely high expression levels overwhelm signals from other genes, and therefore had to be removed from the experiment. This is a limitation of the Molecular Cartography platform. Unfortunately, Gnai2 was determined to be one of these genes and was not evaluated for this purpose.

Canonical OSNs: The authors report a cluster of cells expressing neuronal markers and ORs and call them canonical OSN. However, VSNs expressing ORs have already been reported in a detailed study showing their morphology and location inside the sensory epithelium (References 82, 83). Such cells are not canonical OSNs since they do not show ciliary processes, they express TRPC2 channels and do not express Golf. Are the "canonical OSNs" reported in this study and the OR-expressing VSNs (ref 82, 83) different? Which parameters, other than Gnal and Cnga2 expression, support the authors' bold claim that these are "canonical OSNs"? What is the morphology of these neurons? In addition, the mapping of these "canonical OSNs" shown in Figure 2D paints a picture of the negligible expression/role of these cells (see their prediction confidence).

We observe OR expression in VSNs in our data; these cells cluster with VSNs. The putative mOSN cluster exhibits its own trajectory, distinct from VSN clusters. These cells express Gnal (Golf), which is not expressed in VSNs expressing ORs, nor in any other cell-type in the data. After performing differential gene expression on the putative mOSN cluster, comparing with V1R and V2R VSNs, independently, GO analysis returned the top significantly enriched GO cellular component, ‘cilium’. This new piece of data is presented in the updated Figure S6. Because we were limited to list of 100 genes in Molecular Cartography probe panel, we have prioritized the detection of canonical VNO cell-types, vomeronasal receptor co-expression, and the putative sVSNs, and were not able to include a robust analysis of the putative OSNs.

Secretory VSN: The authors report another novel type of sensory neurons in the VNO and call them "secretory VSNs". Here, the authors performed an analysis of differentially expressed genes for neuronal cells (dataset 2) and found several differentially expressed genes in the sVSN cluster. However, it would be interesting to perform a gene expression analysis using the whole dataset including neuronal and non-neuronal cells. Could the authors find any marker gene that unequivocally identifies this new cell type?

We did not find unequivocal marker genes for sVSNs. We did perform differential analysis of the sVSN cluster with whole VNO data and with the neuronal subset, as well as against specific cell-types. We could not find a single gene that was perfectly exclusive to sVSNs. We used a combinatorial marker-gene approach to predicting sVSNs in the Molecular Cartography data. This required a larger subset of our 100 gene panel to be dedicated to genes for detecting sVSNs.

When the authors evaluated the distribution of sVSN using the Molecular Cartography technique, they found expression of sVSN in both sensory and non-sensory epithelia. How do the authors explain such unexpected expression of sensory neurons in the non-sensory epithelium?

In our scRNA-Seq experiment, blood vessels were removed, limiting the power to distinguish between certain cell types. Because of the limited number of genes that we can probe using Molecular Cartography, the number of genes associated with sVSNs may be present in the non-sensory epithelium. This could lead to the identification of cells that may or may not be identical to the sVSNs in the non-neuronal epithelium. Indeed, further studies will need to be conducted to determine the specificity of these cells.

The low total genes count and low total reads count, combined with an "expression of marker genes for several cell types" could indicate low-quality beads (contamination) that were not excluded with the initial parameter setting. It looks like cells in this cluster express a bit of everything V1R, V2R, OR, secretory proteins.

We are confident that the putative sVSN cell cluster is not the result of low-quality cells. We performed a robust quality-control protocol, including count correction for ambient RNA with the R package, SoupX, multiplet cell detection and removal with the Python module, Scrublet, and a strict 5% mitochondrial gene expression cut-off. Furthermore, the cell clusters in question show no signs of being the result of sequencing artefacts, as they are connected in a reasonable orientation to the rest of the neuronal lineage in modular clusters in 2D and 3D UMAP space. The OSN and sVSN cell clusters each show distinct and self-consistent expressions of genes (Fig. S1H). Gene ontology (GO) analysis reveals significant GO term enrichment for both the sVSN (Fig. 2G) and mOSN clusters when compared to mature V1R and V2R VSNs, indicating functional differences. Moreover, while some genes were expressed at a lower level when compared to the canonical VSNs, others were expressed at higher levels, precluding the cause of discrepancy as resulting from an overall loss of gene counts.

The authors wrote ‘...the transcriptomic landscape that specifies the lineages is not known...’. This statement is not completely true, or at least misleading. There are still many undiscovered aspects of the transcriptomics landscape and lineage determination in VSNs. However, authors cannot ignore previously reported data showing the landscape of neuronal lineages in VSNs (Ref ref 88, 89, 90, 91 and doi.org/10.7554/eLife.77259). Expression of most of the transcription factors reported by this study (Ascl1, Sox2, Neurog1, Neurod1...) were already reported, and for some of them, their role was investigated, during early developmental stages of VSNs (Ref ref 88, 89, 90, 91 and doi.org/10.7554/eLife.77259). In summary, the authors should fully include the findings from previous works (Ref ref 88, 89, 90, 91 and doi.org/10.7554/eLife.77259), clearly state what has been already reported, what is contradictory and what is new when compared with the results from this work.

This is a difference in opinion about the terminology. Transcriptomic landscape in our paper refers to the genome-wide expression by individual cells, not just individual genes. The reviewer is correct that many of the genetic specifiers have been identified, which we cited and discussed. We consider these studies as providing a “genetic” underpinning, rather than the “transcriptomic landscape” in lineage progression. To avoid confusion, we have revised the statement to “… the transcriptional program that specifies the lineages is not known.”

…the co-expression of specific V2Rs with specific transcription factors does not imply a direct implication in receptor selection. Directed experiments to evaluate the VR expression dependent on a specific transcription factor must be performed.

The reviewer is correct, and we did not claim that the co-expression of specific transcription factors indicates a direct relationship with receptor selection. We agree that further directed experiments are required to investigate this question.

This study reports that transcription factors, such as Pou2f1, Atf5, Egr1, or c-Fos could be associated with receptor choice in VSNs. However, no further evidence is shown to support this interaction. Based on these purely correlative data, it is rather bold to propose cascade model(s) of lineage consolidation.

The reviewer is correct. As any transcriptomic study will only be correlative, additional studies will be needed to unequivocally determine the mechanistic link between the transcription factors with receptor choice. Our model provides a basis for these studies.

The authors use spatial molecular imaging to evaluate the co-expression of many chemosensory receptors in single VNO cells. […] However, it is difficult to evaluate and interpret the results due to the lack of cell borders in spatial molecular imaging. The inclusion of cell border delimitation in the reported images (membrane-stained or computer-based) could be tremendously beneficial for the interpretation of the results.

The most common practice for cell segmentation of spatial transcriptomics data is to determine cell borders based on nuclear staining with expansion. We have tested multiple algorithms based on recent studies, but each has its own caveat.

It is surprising that the authors reported a new cell type expressing OR, however, they did not report the expression of ORs in Molecular Cartography technique. Did the authors evaluate the expression of OR using the cartography technique?

We were limited to a 100-gene probe panel and only included one OR. The expression was not high enough for us to substantiate any claims.

**Reviewer #3:**
(1) The authors claim that they have identified two new classes of sensory neurons, one being a class of canonical olfactory sensory neurons (OSNs) within the VNO. This classification as canonical OSNs is based on expression data of neurons lacking the V1R or V2R markers but instead expressing ORs and signal transduction molecules, such as Gnal and Cnga2. Since OR-expressing neurons in the VNO have been previously described in many studies, it remains unclear to me why these OR-expressing cells are considered here a "new class of OSNs." Moreover, morphological features, including the presence of cilia, and functional data demonstrating the recognition of chemosignals by these neurons, are still lacking to classify these cells as OSNs akin to those present in the MOE. While these cells do express canonical markers of OSNs, they also appear to express other VSN-typical markers, such as Gnao1 and Gnai2 (Figure 2B), which are less commonly expressed by OSNs in the MOE. Therefore, it would be more precise to characterize this population as atypical VSNs that express ORs, rather than canonical OSNs.

We observe OR expression in VSNs in our data; these cells cluster with VSNs. The putative mOSN cluster exhibits its own trajectory, distinct from VSN clusters. These cells express Gnal (Golf), which is not expressed in VSNs expressing ORs, nor in any other cell-type in the data. We have performed differential gene expression analysis on the putative mOSN cluster to compare with V1R and V2R VSNs. GO analysis returned the top significantly enriched GO terms, including many related to “cilium”., further supporting that these are OSNs. Because we were limited to list of 100 genes in Molecular Cartography probe panels, we have prioritized the detection of canonical VNO cell-types, vomeronasal receptor co-expression, and the putative sVSNs, and were not able to include a robust analysis of the putative OSNs. With regard to Gnai2 and Go expression, we have examined our data from the OSNs dissociated from the olfactory epithelium and detected substantial expression of both. This new analysis provides additional support for our claim. We now present differentially expressed genes and GO term analysis of the mOSN class in the updated Figure S6.

(2) The second new class of sensory neurons identified corresponds to a group of VSNs expressing prototypical VSN markers (including V1Rs, V2Rs, and ORs), but exhibiting lower ribosomal gene expression. Clustering analysis reveals that this cell group is relatively isolated from V1R- and V2R-expressing clusters, particularly those comprising immature VSNs. The question then arises: where do these cells originate? Considering their fewer overall genes and lower total counts compared to mature VSNs, I wonder if these cells might represent regular VSNs in a later developmental stage, i.e., senescent VSNs. While the secretory cell hypothesis is compelling and supported by solid data, it could also align with a late developmental stage scenario. Further data supporting or excluding these hypotheses would aid in understanding the nature of this new cell cluster, with a comparison between juvenile and adult subjects appearing particularly relevant in this context.

We wholeheartedly agree with this assessment. Our initial thought was that these were senescent VSNs, but the trajectory analysis did not support this scenario, leading us to propose that these are putative secretive cells. Our analysis also shows that overall, 46% of the putative sVSNs were from the P14 sample and 54% from P56. These cells comprise roughly 6.4% of all P14 cells and 8.5% of P56 cells. In comparison, 28.4% of all cells are mature V1R VSNs at P14, but the percentage rise to 46.7% at P56. The significant presence of sVSNs at P14, and the disproportionate increase when compared with mature VSNs indicate that these are unlikely to be late developmental stage or senescent cells, although we cannot exclude these possibilities.

We have included the sVSNs in a trajectory inference analysis and found that the pseudotime values of the sVSNs are within the range of those cells within the V1R and V2R lineages, indicating a similar maturity (Fig. S6).

(3) The authors' decision not to segregate the samples according to sex is understandable, especially considering previous bulk transcriptomic and functional studies supporting this approach. However, many of the highly expressed VR genes identified have been implicated in detecting sex-specific pheromones and triggering dimorphic behavior. It would be intriguing to investigate whether this lack of sex differences in VR expression persists at the single-cell level. Regardless of the outcome, understanding the presence or absence of major dimorphic changes would hold broad interest in the chemosensory field, offering insights into the regulation of dimorphic pheromone-induced behavior. Additionally, it could provide further support for proposed mechanisms of VR receptor choice in VSNs.

The reviewer raised a good point. We did not observe differences between male and female, or between P14 and P56 mice in the distribution of clusters and cells in UMAP space. Indeed, our differential expression analysis has revealed significantly differentially expressed genes in both comparisons. Results from these analyses are presented in the new Figures S1 and S2.

(4) The expression analysis of VRs and ORs seems to have been restricted to the cell clusters associated with the neuronal lineage. Are VRs/ORs expressed in other cell types, i.e. sustentacular, HBC, or other cells?

Sparsely expressed low counts of VR and OR genes were observed in non-neuronal cell-types. When their expression as a percentage of cell-level gene counts is considered, however, the expression is negligible when compared to the neurons. The observed expression may be explained by stochastic base-level expression, or it may be the result of remnant ambient RNA that passed filtering.